# Sarcopenic Obesity and Sarcopenic Visceral Obesity, Calculated Using the Skeletal Muscle İndex and Visceral Fat İndex at the L3 Vertebra Level, Do Not Predict Survival Rates in Endometrial Cancer Patients

**DOI:** 10.3390/jcm14227915

**Published:** 2025-11-07

**Authors:** Melek Özdemir, Gamze Gököz Doğu, Burcu Yapar Taşköylü, Muhammet Arslan, Burak Kurnaz, Atike Gökçen Demiray, Arzu Yaren, Serkan Değirmencioğlu, Yeliz Arman Karakaya

**Affiliations:** 1Medical Oncology Clinic, Denizli State Hospital, 20010 Denizli, Turkey; 2Medical Oncology, Department of Internal Medicine, Pamukkale University, 20160 Denizli, Turkey; ggd2882@gmail.com (G.G.D.); drburcuyapar@gmail.com (B.Y.T.); gokcenakaslan@gmail.com (A.G.D.); arzu_yaren@yahoo.com (A.Y.); 3Department of Radiology, Pamukkale University, 20160 Denizli, Turkey; muhammetarslan@pau.edu.tr (M.A.); burakkurnaz94@gmail.com (B.K.); 4Medical Oncology Clinic, Denipol Hospital, 20010 Denizli, Turkey; drserkandeg@hotmail.com; 5Department of Patology, Pamukkale University, 20160 Denizli, Turkey; ykarakaya@pau.edu.tr

**Keywords:** visceral adipose tissue, survival analysis, cross-sectional imaging, lymphovascular invasion, serous carcinoma

## Abstract

**Objective**: Obesity increases the risk of endometrial cancer (EC). In this study, we aimed to investigate the prognostic effect of sarcopenia, sarcopenic obesity and sarcopenic visceral obesity, calculated with the help of cross-sectional imaging methods of muscle and visceral adipose tissue from body composition parameters, in EC. **Methods**: Patients diagnosed with EC were identified between January 2014 and June 2024. The combination of radiological markers and patient outcomes can predict prognosis. The skeletal muscle index (SMI) and visceral fat index (VFI) were calculated from computed tomography (CT) and/or abdominal magnetic resonance (MR) scans taken at the time of diagnosis at the Lumbal 3 (L3) vertebra level. The findings of these analyses demonstrate the strongest correlation with the ratio of muscle and visceral fat tissue throughout the body. The loss of muscle and fat is an unfavourable indicator in patients with EC. The present study analysed the prognostic values of sarcopenia, sarcopenic obesity, sarcopenic visceral obesity, and the visceral fat index in EC. The total skeletal muscle area was calculated in square centimetres. Body surface area (m^2^) was calculated using the Mosteller formula: ((height (cm) × weight (kg))/3600)^1/2^. To normalize body composition components, the skeletal muscle index was calculated as cm^2^/m^2^. **Results**: The study comprised a total of 236 EC patients. The prevalence of sarcopenia, sarcopenic obesity, and sarcopenic visceral obesity were found to be 48.31%, 33.47%, and 22.88%, respectively. The presence of sarcopenia, high VFI levels, sarcopenic obesity, and sarcopenic visceral obesity did not demonstrate statistical significance in the survival analysis. However, stage increase (*p* = 0.001), primary tumour localization in the lower uterine segment (*p* = 0.001), serous carcinoma (*p* = 0.001), increased grade in endometrioid carcinoma (*p* = 0.023), and lymphovascular invasion (*p* = 0.001) were significantly associated with increased mortality risk. The presence of sarcopenia was found to be significant in patients with obesity (*p* = 0.008) and those aged ≥ 65 years (*p* = 0.001). **Conclusions**: In EC survival, established prognostic factors such as serous histopathology, LVI positivity, and the extent of surgical staging are prioritised. The presence of these well-established markers means the potential effect of BMI-based observations, such as the ‘obesity paradox’, and even body composition measurements, such as sarcopenic obesity, are now statistically insignificant. Our findings suggest that aggressive tumour biology (serous type, LVI) and surgery, rather than metabolic variables such as sarcopenia, sarcopenic obesity and sarcopenic visceral obesity, are the direct reason for the survival difference. This is due to the tumour’s aggressive nature and clinical characteristics (e.g., age at diagnosis, operability, stage, primary tumour localization in the lower uterine segment, serous carcinoma, grade, and LVI positivity) rather than metabolic variables.

## 1. Introduction

Endometrial carcinoma (EC) is the second most prevalent gynaecological cancer worldwide [1]. The most common age at diagnosis is between 60 and 70 years [2]. A review of the literature reveals that patients diagnosed under the age of 50 exhibit a number of high-risk factors, including anovulation, nulliparity, obesity, diabetes mellitus, metabolic syndrome, advanced age, black race and a history of breast cancer [3,4,5,6,7,8].

In the case of EC, the most significant risk factor is identified as high estrogen levels, resulting from endogenous and/or exogenous intake, in conjunction with peripheral fat aromatization induced by obesity [9,10,11,12]. The visceral fat index is a measure of visceral obesity and total fat mass [13]. Sarcopenia is defined as the loss of skeletal muscle mass. The presence of cancer cachexia has been demonstrated to result in a number of adverse outcomes for patients, including a poor prognosis and the development of psychosocial disorders, which can have a significant impact on both the patient and their family [14]. While the majority of cachectic patients are sarcopenic, not all sarcopenic patients are cachectic. The term ‘sarcopenic obesity’ refers to the loss of muscle mass without concomitant fat loss. Immobilization related atrophy, chronic inflammation, insulin resistance, and nutritional deficiencies have been identified as contributing factors to sarcopenic obesity [8,15,16].

In the Global Cancer Statistics 2020 (GLOBOCAN) study, which examined 36 cancer subtypes on a global scale, the number of new cases of endometrial cancer was recorded as 417,367, with an annual mortality rate of 97,370. The incidence of the condition is higher in developed countries. The incidence of the condition is also increasing in Turkey [1]. It has been hypothesised that the combination of radiological markers with patients’ clinical and laboratory results may be useful in predicting prognosis. The skeletal muscle index (SMI) and visceral fat index (VFI) were calculated from measurements taken at the lumbar 3 (L3) vertebra level in abdominal computed tomography (CT) and/or abdominal magnetic resonance (MR) radiological images taken for the purpose of new diagnosis and staging. It has been established that these analyses demonstrate the strongest correlation with the ratio of muscle and visceral fat tissue throughout the body [17]. The loss of muscle tissue and the increase in fat tissue in EC patients are considered to be unfavourable prognostic factors. In the present study, the prognostic values of sarcopenia, sarcopenic obesity, sarcopenic visceral obesity and the visceral fat index in EC were analysed by calculating these radiological markers.

## 2. Materials and Methods

### 2.1. Data Collection and Patient Characteristics

This study is of a retrospectively and cross-sectionally conducted archival nature. Patients aged 18 years or older with a diagnosis of EC who were followed up at the oncology clinic during the specified period (January 2014–June 2024) and who met the exclusion criteria were included in the study. Patients with a secondary malignancy, a history of diabetes mellitus, chronic renal failure, chronic rheumatic disease, or chronic lung disease were excluded from the study. Furthermore, patients for whom no radiological data was available were also subjected to evaluation as exclusion criteria. The absence of a sample size calculation is a significant methodological limitation of this study, particularly given that it is an observational study aiming to identify prognostic relationships.

We also meticulously documented the use of abdominal CT and/or MRI scans for diagnosis and staging. We calculated SMI and VFI from radiological sections taken from the L3 vertebral level, as this has been shown to be the most reliable way to measure the ratio of muscle and visceral adipose tissue in the whole body. All measurements were performed on abdominal CT examinations acquired with a 128-detector array multislice CT scanner (Ingenuity 128, Philips Healthcare, Cleveland, OH, USA). The imaging parameters for the CT scanner were as follows: tube voltage of 100 kV; tube current of 150–200 mAs; slice thickness of 1.5 mm; collimation of 3 × 1.5 mm; matrix size of 512 × 512. The muscle areas of the rectus abdominis, transversus abdominis, internal and external obliques, erector spinae, quadratus lumborum and psoas were calculated. The Hounsfield unit (HU) is a measure of muscle tissue, with values ranging from −29 HU to +150 HU, and visceral adipose tissue, ranging from −150 HU to −50 HU. Visceral and subcutaneous adipose tissue were distinguished manually, using the abdominal wall and paraspinal muscles as anatomical boundaries.

The measurement of these radiologic markers was conducted on two separate occasions, with the same investigator performing the assessments. The measurements were then subjected to a second inspection by the second investigator. Consequently, the results of all patients were evaluated using the same standards. The cut-off value of Body Mass Index (BMI) (kg/m^2^) obtained from studies in the literature was used to group these measurements without knowing the clinical and laboratory data of the patients. Body surface area (m^2^) was calculated using Mosteller’s formula: ((height (cm) × weight (kg))/3600))^1/2^. Skeletal muscle area was measured in cm^2^. The skeletal muscle index (cm^2^/m^2^) was calculated by normalising body composition components [16].
Body mass index; BMI < 25 normal Weight, BMI: 25–29 Overweight, BMI ≥ 30 Obesity.Sarcopenia was calculated as skeletal muscle area divided by height squared (cm^2^/m^2^) at L3 vertebral level. Since there was no standardized cut-off value, the cut-off value was accepted as the median value. Calculated measurements below the median value were considered as sarcopenia.VFI was calculated as the median value due to the lack of standardized cut-off values for VFI. The cohort was divided into low and high VFI.Low SMI = sarcopenia,High BMI + Low SMI = sarcopenic obesity,High VFI + Low SMI = sarcopenic visceral obesity [16].

The effect of these prognostic markers on survival in EC was evaluated. Clinical and demographic data were retrospectively recorded from the patients’ files at the time of diagnosis. The analysis encompassed various metrics, including age at diagnosis, stage, systemic treatment, treatment responses, progression, and survival data. The primary outcomes of interest were median overall survival (mOS).

### 2.2. Statistical Analysis

The descriptive statistics of the data obtained from the study included the mean, standard deviation, median, minimum and maximum value for numerical variables, and frequency and percentage analysis for categorical variables. An investigation was conducted into the conformity of the numerical variables to the normal distribution, with the Shapiro–Wilk test being utilised for this purpose. Independent samples *t*-tests or Mann–Whitney U tests were used to compare these variables according to categorical variables. Furthermore, differences between categorical variables were subjected to Chi-square analysis. The Kaplan–Meier test was utilised for the purpose of survival analysis. Furthermore, univariate and multivariate Cox regression analysis was employed to evaluate variables that may have an effect on survival. mOS was defined as the time from the date of diagnosis to the time of death or last follow-up. The analyses were conducted utilising the SPSS 22.0 programme, with a significance level of *p* < 0.05 being selected.

## 3. Results

During the specified time interval, 244 patients were diagnosed with EC. A comprehensive analysis was conducted on 236 patients, encompassing complete imaging data. Accordingly, 48.31% of patients exhibited sarcopenia, 33.47% exhibited sarcopenic obesity, and 22.88% exhibited sarcopenic visceral obesity. The VFI value was elevated in 50.42% of cases. The study revealed that 60.59% of the patients were obese. The age at diagnosis was found to be less than 65 years in 62.71% of cases, with a median age of 61.06 ± 10.18 (53.95–68.81) years. The clinical and demographic characteristics of the patients are set out in Table 1.

The clinical and demographic data of the grouping according to sarcopenia and VFI status in the subgroup analysis are presented in Table 2. The findings indicated that the prevalence of obesity and Proficient mismatch repair (pMMR) gene was elevated in sarcopenic patients (*p* < 0.05). Patients with low Eastern Cooperative Oncology Group (ECOG) Performance Status were more prevalent among sarcopenic patients. Although this did not reach statistical significance, the VFI value was higher in patients aged ≥65 years who were obese (see Table 2).

The mOS of the patients was determined as 85.341 ± 3.775 months. Sarcopenia was not a predictive prognostic marker for mOS (mOS = 82.347 ± 5.504; mOS = 87.947 ± 5.119 months, *p* > 0.05) (Figure 1). The life expectancy of patients with VFI (mOS = 87.947 ± 5.119; mOS = 82.347 ± 5.504, *p* > 0.05, Figure 1), Sarcopenic obesity (mOS = 85.394 ± 4.584; mOS = 83.868 ± 6.404, *p* > 0.05, Figure 2) and Sarcopenic visceral obesity (mOS = 85.022 ± 4.242; mOS = 84.645 ± 8.105, *p* > 0.05, Figure 2) classification did not show a significant relationship with mOS.

Univariate and multivariate Cox regression analysis was conducted, revealing that age at diagnosis, metastasis status and surgery had a significant effect on survival (*p* < 0.05). The mortality rate increased twofold (*p* = 0.001) when the age at diagnosis exceeded 65 years. In accordance with the extant literature, mortality risk increased with increasing stage (*p* < 0.05). The primary tumour’s location in the lower uterine segment was found to increase mortality by 2.4 times (*p* = 0.001). The mortality rate of patients diagnosed with serous carcinoma histopathology was found to be 3.1 times higher than that of patients diagnosed with endometrioid carcinoma (*p* = 0.001). In comparison with endometriod carcinoma grade 1, the mortality rate was 1.3 times higher in patients diagnosed with grade 2 disease, and 2.4 times higher in those diagnosed with grade 3 disease (*p* = 0.023). In instances where lymphovascular invasion was present in histopathological analysis, the mortality rate increased threefold (*p* = 0.001), as illustrated in Table 3.

## 4. Discussion

The present study was conducted with the objective of investigating the prognostic impact of sarcopenia, sarcopenic obesity and sarcopenic visceral obesity in EC. Contrary to prevailing expectations, the presence of sarcopenia, VFI level, sarcopenic obesity and sarcopenic visceral obesity were not found to be significantly associated with patient survival. Univariate and multivariate Cox regression analysis of clinical and demographic characteristics demonstrated that age at diagnosis of less than 65 years, operable tumour and absence of metastasis were significantly associated with survival. Increased stage, lower uterine segment localization of the primary tumour, serous carcinoma, increased grade in endometriod carcinoma and LVI were significantly associated with increased mortality risk.

The utilisation of BMI as a solitary metric to ascertain the nutritional status of patients is inadequate for this purpose [18]. Consequently, the necessity for novel prognostic markers is being considered. In contrast to BMI, SMA and VFI provide insight into muscle, fat and protein metabolism. Despite the absence of a substantial correlation between BMI and survival in the present study (*p* = 0.932), obesity remains a pivotal factor contributing to an elevated risk of EC [12,19,20,21]. The elevated risk of EC in individuals with obesity can be mitigated by regular physical activity [22].

As demonstrated by a number of studies [23,24,25], physical activity has been shown to have a positive influence on the health of cancer patients, offering significant benefits. These include the preservation of muscle mass, the reduction in central adiposity, the acceleration of metabolism, the regulation of metabolic hormone levels and the decrease in insulin resistance.

In the present study, it was demonstrated that mortality increased 2.5-fold (*p* = 0.001) in patients over the age of 65 at the time of diagnosis. This finding suggests that the observed increase in mortality may be attributable to accelerated muscle loss associated with advancing age in adults. Sarcopenia is an independent poor prognostic factor in EC patients, who are mostly diagnosed at an advanced age [26]. The loss of muscle mass can be averted by increasing levels of physical activity. Consequently, EC should be regarded as a public health concern, and physical activity should be advocated as a preventative health measure to mitigate the risk of EC development.

The equilibrium between weight gain and weight loss is of paramount importance. On the one hand, patients are advised to maintain a healthy weight; on the other hand, weight loss is an undesirable outcome. The distinction between cancer cachexia and weight loss is the disproportionate loss of lean body mass. The condition has been observed to induce psychosocial disorders within the family unit and to portend a poor prognosis for the patient [14].

The term “obesity paradox” has been the subject of debate in studies in the literature, but its precise meaning remains unclear. Consequently, the findings of the analysis presented in extant studies remain speculative. The question of whether this defined paradox is the result of methodological shortcomings or a real biological protection remains a subject of debate. Although obesity is a significant adverse risk factor in the general population, its reversal in patients with a history of chronic disease creates the so-called “obesity paradox”. The potential rationales underpinning its speculative character have been identified. The preponderance of this phenomenon has been attributed to methodological deficiencies and statistical artefacts. It is argued that reverse causality is the most compelling argument [27].

The inflammatory process, which escalates concomitantly with the progression of chronic disease, engenders a condition known as “disease-related cachexia.” This phenomenon is known to result in unintentional weight loss. Patients who appear to be of normal weight despite significant weight loss may in fact be in the advanced stages of the disease, and their apparent normality is a consequence of loss of muscle and fat tissue. In summary, obesity does not offer protection; it has been observed that individuals may exhibit normal or thin body mass because of the progression of the disease. Therefore, it may be posited that a reduction in thickness is indicative of an exacerbation of the disease [27,28].

Another factor attributed to the obesity paradox is the inadequacy of BMI as a comprehensive assessment tool for body composition. It is important to note that BMI provides only approximate data and does not make any comment on muscle mass and fat mass [29]. The patient may be sarcopenic, characterized by a loss of muscle mass, despite having a BMI result that places them in the obese category. Conversely, patients who, despite being within the normal weight BMI range, exhibit signs of muscle loss may be categorized as frail. The findings of this study demonstrate that BMI is not a reliable indicator [30].

Another view argues that the biological basis of the obesity paradox is the Energy Reserve or Hibernation Hypothesis. The hypercatabolic process caused by chronic diseases increases the body’s energy requirements. It is thought that obesity, accompanied by the preservation of normal muscle tissue, enables the body to cope with catabolic stress. Fragile patients, however, cannot cope with the increased hypercatabolic state [31,32]. This theory is consistent with evolutionary theory. The hypothesis underpinning this theory is that species utilize fat tissue stored for periods of starvation during hibernation [33].

Finally, it was considered that the prognostic variables evaluated in the studies could have a confounding effect (Confounding Variables). Smoking has been identified as a poor prognostic factor and is associated with low BMI. In instances where underweight status is associated with smoking, it is observed that the prognosis is unfavorable, despite the poor prognosis. In non-smokers, the obesity paradox was not observed [27,28].

In the present study, 60% (n:71) of patients with high VFI at diagnosis were stage 1, and 78% (n:92) of primary tumors were in the uterine corpus. These results lend support to the notion of the obesity paradox, as they suggest that obesity is associated with more favorable prognostic features. In obese patients, type 1 endometrial carcinoma is more prevalent than aggressive type 2 endometrial cancer. This finding lends further support to the observation that less aggressive tumors are observed in obese patients compared to normal-weight patients [34].

Most of the criticism in studies evaluating the obesity paradox is attributable to the fact that BMI fails to provide discriminatory data on the ratio of muscle to fat tissue. The present study was conducted with the objective of evaluating the prognostic significance of sarcopenic obesity (characterized by low muscle mass and high fat mass). This investigation was undertaken to address a critical aspect of the obesity paradox.

The grounds on which the “obesity paradox” is regarded as “speculative” pertain to reverse causality, the inadequacy of BMI, the energy reserve or hibernation hypothesis, and confounding variables. It is hypothesized that the development of more advanced analyses in this field will necessitate the concurrent measurement of waist circumference, VFI, bioimpedance analysis, and sarcopenia in conjunction with BMI. Conversely, adhering to the “first, do no harm” hypothesis, it is believed that an oncologist’s evaluation of the obesity paradox, considering the methodological limitations previously mentioned, will prevent misinterpretations

There has been an increase in the number of studies evaluating sarcopenia in cancer patients. The administration of high doses of treatment protocols to sarcopenic patients has been demonstrated to exert a detrimental effect on survival outcomes. Consequently, the assessment of sarcopenia is imperative [35,36].

As the stage of the disease increases, sarcopenia also increases. They attributed this to the long palliative care period of patients due to relapses. Postoperative complications, infection and chemotherapy toxicity increase in sarcopenic patients. Therefore, treatment compliance becomes difficult and treatment non-response increases [37,38]. In this study, the mortality risk increased as the stage increased in accordance with the literature (*p* < 0.05). However, stage increase was not associated with histopathologic subtype, grade, LVI and sarcopenia. Of the 79 patients with sarcopenia, 79 (69.3%) were obese and as a result, it was thought that physical mobility restriction due to sarcopenia may have increased obesity. Although a study in the literature concluded that mortality risk increased with sarcopenia [39], in our study, there was no significant difference between obesity and sarcopenia for survival in cox regression analysis. There was no association between sarcopenia and pathologic prognostic variables. Stage, primary tumor localized in the lower uterine segment, histopathology, grade and LVI status were prognostic variables for survival. Consistent with the literature, mortality risk increased with increasing stage (*p* < 0.05). It was observed that sarcopenia was more common in obese patients and those over 65 years of age at diagnosis, and the functional capacity of patients with sarcopenia was reduced.

The coexistence of obesity and sarcopenia is known as sarcopenic obesity. This is suboptimal prognostication, as it increases the risks of both obesity and muscle loss concurrently. The administration of cytotoxic therapy can result in an inaccurate calculation of the required dose, which can lead to a loss of drug efficacy due to the limited volume of drug distributed to the lean body part. Consequently, the prognostic value of this marker has been investigated in various malignancies in the extant literature [40].

In the extant literature, sarcopenic obesity has been demonstrated to be associated with OS in patients with EC, while sarcopenia has not been shown to be associated with OS. In a subgroup analysis, it was concluded that sarcopenic obesity predicted OS only in patients with endometrioid EC [41]. In another EC study, survival of patients with advanced disease who underwent pelvic exenteration was found to be associated with sarcopenia and malnutrition levels. Consequently, the prevailing hypothesis suggests that the concomitant presence of stage and sarcopenia may serve as a prognostic indicator for survival in EC [42].

In a Canadian cancer centre, sarcopenic obesity was associated with functional capacity, OS and chemotherapy-related toxicity in a study involving solid tumours of the lung and gastrointestinal tract. Given the accessibility of CT images for all patients, it has been recommended that body composition should be examined prior to and during the treatment plan, and that drug dosing should be performed accordingly [40].

It is important to note that body weight measurement may underestimate the frequency of sarcopenia in obese patients. Consequently, the calculation of muscle index and fat index derived from radiological measurements using CT/MR will yield more objective results. Research has demonstrated that the anticipated mOS for patients diagnosed with sarcopenia and sarcopenic obesity is eight months, while the survival rate for patients without sarcopenia is 28 months. This study of the literature has demonstrated that sarcopenia and sarcopenic obesity have a high prognostic value (*p* < 0.001). In consideration of the findings, research endeavours aimed at ascertaining whether augmenting caloric intake and administering appetite-stimulating treatments in patients experiencing cancer-related weight loss would result in enhanced survival outcomes have not yielded favourable outcomes. This phenomenon has been attributed to muscle loss that cannot be compensated for by weight gain alone [43,44,45].

There are positive studies as well as negative studies. In this study, although the mOS (83.87 months) of patients with sarcopenic obesity was numerically lower than the mOS (85.40 months) of patients without sarcopenic obesity, it did not reach statistical significance. Similarly to the results of this study, there are studies in the literature in which visceral adipose tissue and muscle tissue compartments do not predict survival. In the study conducted in pancreatic cancer patients, it was thought that the results of muscle and adipose tissue changes caused by a catabolic process would not have been reflected in the clinic because the clinical course of the patients went with rapid progression and short survival data [46].

In our study, we aimed to gain a better understanding of the prognostic impact of metabolic and radiological variables, as well as the well-known strong prognostic factors, in EC. A very recent meta-analysis examining the importance of surgical intervention among prognostic factors indicates that surgery involving lymph node dissection improves survival, particularly for high-risk EC patients (high grade, serous type, LVI positivity). (HR: 0.62, 95% CI: 0.43–0.91). This finding is consistent with the literature and shows that surgery is a key factor in determining survival in high-risk tumors [47].

Another study demonstrated that the surgical approach affects progression-free survival. The effect of open surgery and minimally invasive surgery on survival was analyzed in serous carcinoma, an aggressive subtype. Although 5-year PFS was lower with minimally invasive surgery (49.7–68.3%, *p* = 0.017) and recurrence rates were higher (49.1–31.7%), OS was like that with open surgery. Despite being frequently used in oncological surgical practice, the oncological safety of minimally invasive surgery remains uncertain. Additionally, adjuvant chemotherapy was found to increase PFS (HR = 0.28, 95%CI:0.13–0.60, *p* = 0.001). In conclusion, while minimally invasive surgery reduced PFS in high-risk pathological subtypes in this study, the critical importance of adjuvant therapy in improving survival rates was emphasized once again [48].

In the present study, 222 patients (94.47%) underwent surgical intervention. Surgery was found to be a strong and independent predictor of survival, consistent with the literature (HR: 0.198, 95% CI: 0.063–0.621, *p* = 0.005). A range of other prognostic variables were obtained, including stage (*p* = 0.001), LVI positivity (*p* = 0.001), age ≥ 65 (*p* = 0.001), grade increase (*p* = 0.023), lower uterine segment involvement (*p* = 0.001), presence of metastasis (*p* = 0.001), and serous histopathology (*p* = 0.001). Grade is one of the most fundamental pathological prognostic factors. An increase in grade indicates greater aggressiveness and a poorer overall survival (OS) rate, regardless of pathological subtype (endometrioid or non-endometrioid) [49].

According to the principles of endometrioid pathology, non-endometrioid pathological subtypes, including serous carcinoma, are uniformly poor prognoses, irrespective of stage. Serious carcinoma is frequently linked to p53 mutations, and the 5-year overall survival (OS) expectation is low (Stage I–II: 74%, Stage III–IV: 33%). Despite representing a mere 10% of all ECs, it is responsible for 39–40% of EC-related mortality [50]. The primary treatment for newly diagnosed serous carcinoma is a combination of surgery, chemotherapy, and radiotherapy. Serious carcinoma has been identified as a pathological risk factor which is independently associated with a reduced probability of survival. A review of the literature reveals a correlation between the phenomenon under investigation and both higher recurrence and poorer prognosis, even when comparisons are made on a stage-by-stage basis. Even though this subject has been the focus of research by various study groups worldwide, survival outcomes continue to be poor [51].

The pathological risk factor indicating the spread of cancer cells to blood and lymphatic vessels is lymphovascular invasion (LVI). This is a pivotal step in the process of metastasis. It is a salient pathological feature of tumor aggressiveness. The association between the presence of the condition and an increased risk of recurrence (14.5–6.5%, *p* = 0.026), as well as a poor prognosis in early-stage disease, has been well documented. A study found that LVI increases the risk of recurrence and reduces survival in patients with early-stage EC and negative lymph nodes. This supports LVI in treatment decisions and risk classification [52].

A substantial body of research has demonstrated that Grade [49], serous histology [50,51], and LVI [52] have a significant impact on survival and recurrence, independent of Stage and other clinical factors. LVI and high Grade are regarded as the most critical and independent prognostic markers utilized in adjuvant treatment decisions for endometrial cancer. The prognostic impact of these aggressive histopathological markers is so strong that the additional prognostic value of the metabolic/radiological parameters (sarcopenic obesity and sarcopenic visceral obesity) examined in this study was not found to be statistically significant.

The principal limitation of this study is that it is based on a single-centre, retrospective data set. Due to the retrospective nature of the data analysis, the presence of uncontrollable confounding factors that could affect sarcopenia and visceral adipose tissue measurements could not be fully investigated. A further significant constraint pertains to the temporal arrangement of the CT scans employed for the purpose of evaluating body composition. Delays in the provision of these scans may be attributable to a number of factors, including illness, treatment processes, or patient/physician requests, which have the potential to introduce inconsistencies in the measurements. The absence of SMI or VFI threshold values is a significant limitation and explains our rationale for using the median (i.e., the lack of a population-specific standard). This methodological approach also prevents a full assessment of the potential for “reverse causality” observed in phenomena such as the “obesity paradox” or biases arising from the inadequacies of BMI as a measurement tool. It is the contention of the present study that the measurement of the prognostic value of sarcopenia, sarcopenic obesity and sarcopenic visceral obesity may be enhanced through the execution of prospective, multicentre studies that take sequential measurements during treatment follow-up and address these limitations.

## 5. Conclusions

In conclusion, well-established prognostic factors such as serous histopathology, LVI positivity, and the extent of surgical staging are prioritized in EC survival. The presence of these well-established prognostic markers has rendered the potential prognostic effect of BMI-based observations, such as the “obesity paradox,” and even body composition measurements, such as sarcopenic obesity, statistically insignificant. The findings of this study lend support to the hypothesis that aggressive tumor biology (serous type, LVI) and surgery, rather than metabolic variables such as sarcopenia, sarcopenic obesity, and sarcopenic visceral obesity, are directly responsible for the observed differences in survival. This is due to the tumor’s aggressive histopathology and clinical characteristics (age at diagnosis, operability, stage, primary tumor localization in the lower uterine segment, serous carcinoma, grade, and LVI positivity) being prognostic. It is imperative to methodically evaluate the limitations of the research approach, including the potential for reverse causality in the obesity paradox, the inadequacy of BMI as a measure, the energy reserve hypothesis, and the hibernation hypothesis. Additionally, it is essential to consider confounding variables, which can lead to misinterpretation of the results. Consecutive measurements taken during treatment follow-up in prospective, multicenter studies planned on sarcopenia, sarcopenic obesity, and sarcopenic visceral obesity will be more valuable for measuring prognostic value.

## Figures and Tables

**Figure 1 jcm-14-07915-f001:**
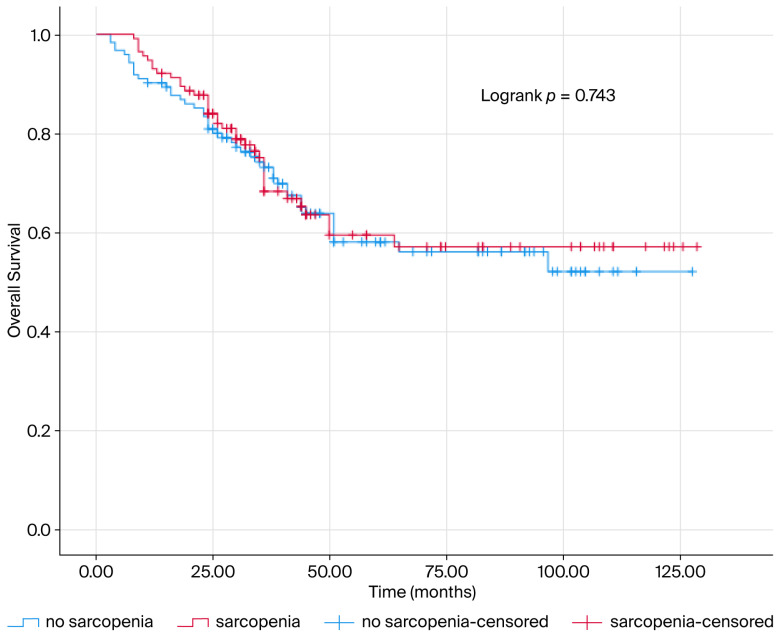
The survival curve is presented according to sarcopenia status. The blue line indicates survival of patients without sarcopenia, while the red line indicates survival of patients with sarcopenia. The Kaplan–Meier estimate of overall survival is presented as a percentage. Sarcopenia and sarcopenic obesity were not found to be prognostic for survival in patients.

**Figure 2 jcm-14-07915-f002:**
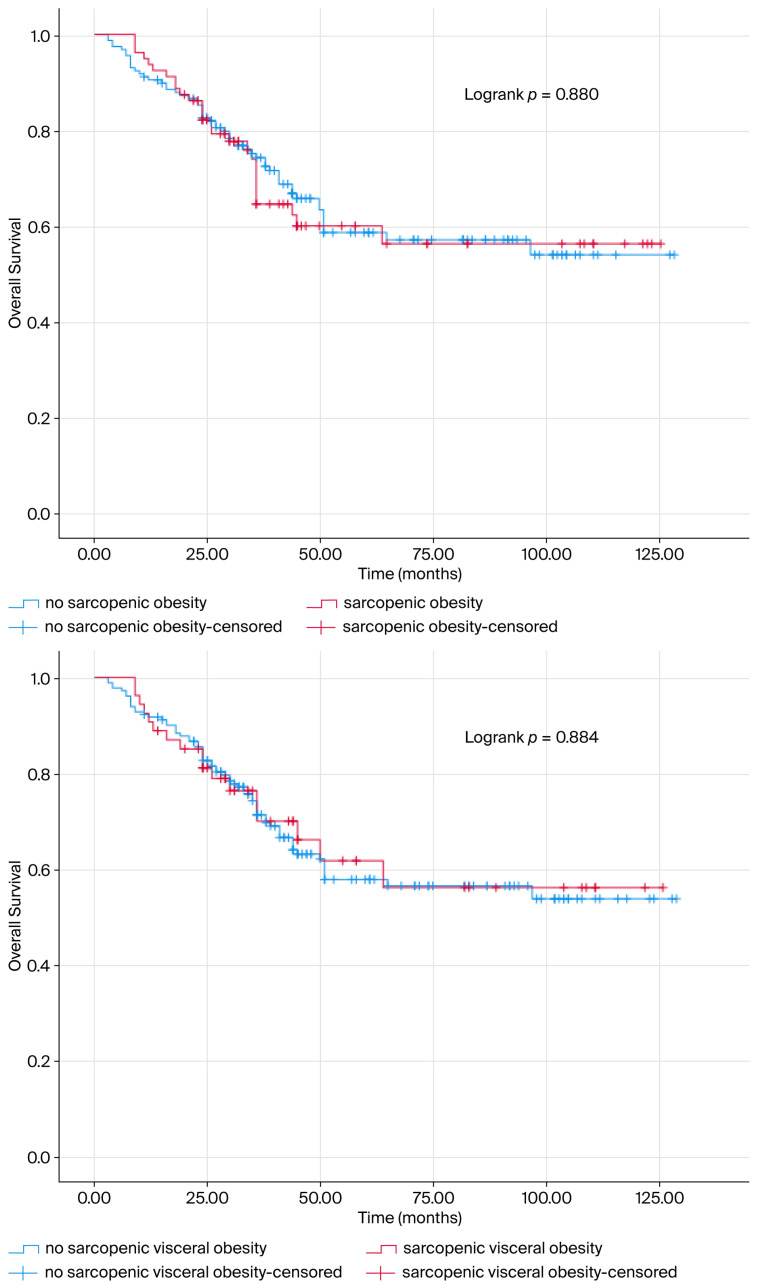
The survival curve is presented according to sarcopenia obesity and sarcopenic visceral obesity status. The blue line indicates survival of patients without sarcopenia obesity and sarcopenic visceral obesity, while the red line indicates survival of patients with sarcopenia obesity and sarcopenic visceral obesity. The Kaplan–Meier estimate of overall survival is presented as a percentage. Sarcopenia obesity and sarcopenic visceral obesity were not found to be prognostic for survival in patients.

**Table 1 jcm-14-07915-t001:** Clinical and demographic characteristics of all patients in the study.

Variables	Numbers (%)	
Sarcopenia	−	122 (51.69)
	+	114 (48.31)
Visceral Fat Index (VFI)	Low	117 (49.58)
	High	119 (50.42)
Obesity	−	93 (39.41)
	+	143 (60.59)
Sarcopenic Obesity	−	157 (66.53)
	+	79 (33.47)
Sarcopenic Visceral Obesity	−	182 (77.12)
	+	54 (22.88)
Age at Diagnosis	<65 Years	148 (62.71)
	≥65 Years	88 (37.29)
Age at Diagnosis (median ± SS)	61.06 ± 10.18	61.15 (53.95–68.81)
Body Mass Index (median ± SS)	31.55 ± 6.38	31.25 (27.56–35.16)
History of Smoking	absence	234 (99.57)
	<43 Packet/year	1 (0.43)
Eastern Cooperative Oncology Group (ECOG) Performance Status	0	183 (77.87)
	1	48 (20.43)
	2	2 (0.85)
	4	2 (0.85)
Histopathology	Endometrioid Carcinoma	194 (82.55)
	Serous Carcinoma	30 (12.77)
	Mixed	4 (1.7)
	Clear Cell Carcinoma	3 (1.28)
	Undifferentiated Carcinoma	2 (0.85)
	High Grade Mucinous Carcinoma	2 (0.85)
Endometrioid Carcinoma	Grade 1	38 (17.27)
	Grade 2	100 (45.45)
	Grade 3	82 (37.27)
Stage	Stage 1	127 (54.04)
	Stage 2	20 (8.51)
	Stage 3	44 (18.72)
	Stage 4	44 (18.72)
Miss Match Repair gene (MMR)	Proficient mismatch repair (pMMR) gene	113 (47.88)
	Deficient mismatch repair (dMMR) gene	24 (10.17)
	unknown	99 (41.95)
Loss of phosphatase and tensin homolog (PTEN) gene	(−) negative	21 (8.9)
	(+) positive	63 (26.69)
	unknown	152 (64.41)
Estrogen Receptor (ER)	unknown	63 (26.69)
	Positive	173 (73.31)
Progesterone Receptor (PR)	Positive	236 (100)
HER2 Score	0	151 (63.98)
	(+) weak positive	13 (5.51)
	(++) moderate positive	8 (3.39)
	(++++) strong positive	12 (5.08)
	unknown	52 (22.03)
P53 Mutation	Positive	146 (61.86)
	unknown	90 (38.14)
Lymphovascular invasion (LVI)	−	146 (62.13)
	+	89 (37.87)
Primary tumor localization	Uterine corpus	175 (74.47)
	Lower Uterine Segment	60 (25.53)
Metastasis	−	138 (75.41)
	+	45 (24.59)
Time to Metastasis	Denova metastasis	49 (51.04)
	Metachronous metastasis	47 (48.96)
Location of metastasis	Lung	23 (24.21)
	Bone	11 (11.58)
	Liver	14 (14.74)
	Adrenal	2 (2.11)
	Peritoneum	33 (34.74)
	Distant lymph node	12 (12.63)
Primer Surgery	−	13 (5.53)
	+	222 (94.47)
Pelvic Radiotherapy	−	99 (42.13)
	+	136 (57.87)
Brachytherapy	−	198 (84.26)
	+	37 (15.74)
Adjuvant Therapy	−	129 (55.13)
	+	105 (44.87)
Adjuvant treatment protocol	Carboplatin + Paclitaxel	81 (77.14)
	others	24 (22.86)
Local Treatment	−	225 (95.74)
	Metastazektomi	4 (1.7)
	Akciğer Radyoterapi	1 (0.43)
	Kemik Radyoterapi	5 (2.13)
First Line Treatment	Cisplatin + Gemcitabine	11 (12.09)
	Carboplatin + Paclitaxel	65 (71.43)
	Cisplatin + Gemcitabine +Bevacizumab	3 (3.3)
	none	7 (7.69)
	others	5 (5.49)
Progression	−	143 (60.59)
	+	93 (39.41)
Living Situation	Alive	154 (65.25)
	Exitus	82 (34.75)

(−): negative, (+): positive.

**Table 2 jcm-14-07915-t002:** Comparison of clinical and demographic variables according to Sarcopenia and Visceral fat index.

Variables		Sarcopenia		*p*	Visceral Fat Index (VFI)		*p*
		Normal	Sarcopenic		Low	High	
		N (%)	N (%)		N (%)	N (%)	
Age at Diagnosis	<65 Years	82 (67.21)	66 (57.89)	0.139	75 (64.1)	73(61.34)	0.661
	≥65 Years	40 (32.79)	48 (42.11)		42 (35.9)	46 (38.66)	
Obesity	no obesity	58 (47.54)	35 (30.7)	0.008 *	50 (42.74)	43 (36.13)	0.299
	Obesity	64 (52.46)	79 (69.3)		67 (57.26)	76 (63.87)	
Eastern Cooperative Oncology Group (ECOG) Performance Status	0	99 (81.15)	84 (73.68)	0.101	91(77.78)	92(77.31)	0.919
	1	19 (15.57)	29 (25.44)		23(19.66)	25(21.01)	
	2	4 (3.28)	1 (0.88)		3(2.56)	2(1.68)	
Histopathology	Endometrioid Carcinoma	102(83.61)	92 (81.42)	0.161	99(84.62)	95(80.51)	0.220
	Serous Carcinoma	12 (9.84)	18 (15.93)		11 (9.4)	19 (16.1)	
	Others	8 (6.56)	3 (2.65)		7 (5.98)	4 (3.39)	
Endometrioid Carcinoma	Grade 1	17 (15.04)	21 (19.63)	0.417	20 (18.02)	18 (16.51)	0.642
	Grade 2	56 (49.56)	44 (41.12)		47 (42.34)	53 (48.62)	
	Grade 3	40 (35.4)	42 (39.25)		44 (39.64)	38 (34.86)	
Stage	Stage 1	66 (54.1)	61 (53.98)	0.417	56 (47.86)	71 (60.17)	0.184
	Stage 2	7 (5.74)	13 (11.5)		12 (10.26)	8 (6.78)	
	Stage 3	25 (20.49)	19 (16.81)		27 (23.08)	17 (14.41)	
	Stage 4	24 (19.67)	20 (17.7)		22 (18.8)	22 (18.64)	
MMR (Miss match repair gen)	Proficient mismatch repair (pMMR) gene	49 (40.16)	64 (56.14)	0.042 *	54 (46.15)	59 (49.58)	0.505
	Deficient mismatch repair (dMMR) gene	13 (10.66)	11 (9.65)		10 (8.55)	14 (11.76)	
	unknown	60 (49.18)	39 (34.21)		53 (45.3)	46 (38.66)	
Loss of phosphatase and tensin homolog (PTEN) gene	−	11 (9.02)	10 (8.77)	0.253	9 (7.69)	12 (10.08)	0.447
	+	27 (22.13)	36 (31.58)		28 (23.93)	35 (29.41)	
	unknown	84 (68.85)	68 (59.65)		80 (68.38)	72 (60.5)	
Estrogen Receptor (ER)	unknown	29 (23.77)	34 (29.82)	0.293	34 (29.06)	29 (24.37)	0.415
	+	93 (76.23)	80 (70.18)		83 (70.94)	90 (75.63)	
P53 Mutation	+	84 (68.85)	62 (54.39)	0.022 *	67 (57.26)	79 (66.39)	0.149
	unknown	38 (31.15)	52 (45.61)		50 (42.74)	40 (33.61)	
Lymphovascular invasion (LVI)	−	75 (61.48)	71 (62.83)	0.830	68 (58.12)	78 (66.1)	0.207
	+	47 (38.52)	42 (37.17)		49 (41.88)	40 (33.9)	
Primary tumor localization	Uterine corpus	91 (74.59)	84 (74.34)	0.964	83 (70.94)	92 (77.97)	0.217
	Lower Uterine Segment	31 (25.41)	29 (25.66)		34 (29.06)	26 (22.03)	
Metastasis	−	71 (74.74)	67 (76.14)	0.826	67 (73.63)	71 (77.17)	0.577
	+	24 (25.26)	21 (23.86)		24 (26.37)	21 (22.83)	
Surgery	−	8 (6.56)	5 (4.42)	0.475	7 (5.98)	6 (5.08)	0.763
	+	114(93.44)	108 (95.58)		110(94.02)	112(94.92)	
Pelvic Radiotherapy	−	55 (45.08)	44 (38.94)	0.341	45 (38.46)	54 (45.76)	0.257
	+	67 (54.92)	69 (61.06)		72 (61.54)	64 (54.24)	
Brachytherapy	−	100(81.97)	98 (86.73)	0.317	96 (82.05)	102(86.44)	0.356
	+	22(18.03)	15 (13.27)		21 (17.95)	16 (13.56)	
Adjuvant Therapy	−	69(56.56)	60 (53.57)	0.646	58 (49.57)	71 (60.68)	0.088
	+	53(43.44)	52 (46.43)		59 (50.43)	46 (39.32)	
Progression	−	69(56.56)	74(64.91)	0.189	73(62.39)	70(58.82)	0.575
	+	53(43.44)	40(35.09)		44(37.61)	49 (41.18)	

(−): negative, (+): positive. * *p* < 0.05. Chi-square test, independent samples; *t* test, Mann–Whitney U test.

**Table 3 jcm-14-07915-t003:** Univariate and Multivariate Cox regression analysis of prognostic variables.

Variables		Univariate Analysis		Multivariate Analysis	
		HR (95% CI)	*p*	HR (95% CI)	*p*
Sarcopenia	No sarcopenia	1 (reference)			
	Sarcopenia	1.075 (0.695–1.662)	0.745		
Visceral fat index (VFI)	Low	1 (reference)			
	High	1.219 (0.790–1.881)	0.371		
Sarcopenic obesity	−	1 (reference)			
	+	1.036 (0.653–1.643)	0.881		
Sarcopenic visceral obesity	−	1 (reference)			
	+	0.961 (0.563–1.640)	0.885		
Age at Diagnosis	<65 Years	1 (reference)		1 (reference)	
	≥65 Years	2.590 (1.672–4.014)	0.001 *	2.882 (1.459–5.692)	0.002 *
Obesity	−	1 (reference)			
	Obesity	1.046 (0.671–1.631)	0.844		
Eastern Cooperative Oncology Group (ECOG) Performance Status	0	1 (reference)	0.173		
	1	1.588 (0.972–2.594)	0.065		
	2	1.436 (0.350–5.887)	0.615		
Histopathology	Endometrioid Carcinoma	1 (reference)	0.001 *	1 (reference)	0.084
	Serous Carcinoma	3.129 (1.799–5.442)	0.001 *	2.226(0.945–5.242)	0.067
	Others	2.343 (1.009–5.441)	0.048 *	3.178 (0.686–14.734)	0.140
Endometrioid Carcinoma	Grade 1	1 (reference)	0.021 *	1 (reference)	0.095
	Grade 2	1.396 (0.636–3.062)	0.406	1.685 (0.575–4.940)	0.342
	Grade 3	2.438 (1.130–5.259)	0.023 *	0.728 (0.234–2.262)	0.583
Stage	Stage 1	1 (reference)	0.001 *	1 (reference)	0.519
	Stage 2	3.183 (1.413–7.170)	0.005 *	0.904 (0.250–3.274)	0.878
	Stage 3	2.728 (1.500–4.962)	0.001 *	1.421 (0.600–3.364)	0.424
	Stage 4	8.886 (5.086–15.528)	0.001 *	1.932 (0.698–5.343)	0.205
Miss match repair gene (MMR)	Proficient mismatch repair (pMMR) gene	0.836 (0.526–1.330)	0.743		
	Deficient mismatch repair (dMMR) gene	0.868 (0.389–1.937)	0.450		
	Unknown	1 (reference)	0.743		
Loss of phosphatase and tensin homolog (PTEN) gene	−	0.742 (0.298–1.846)	0.521	0.378 (0.086–1.671)	0.200
	+	0.413 (0.204–0.833)	0.014 *	0.364 (0.130–1.018)	0.054
	Unknown	1 (reference)	0.044 *	1 (reference)	0.084
Estrogen Receptor (ER)	Unknown	1 (reference)			
	+	0.711 (0.450–1.122)	0.143		
P53 Mutation	+	0.785 (0.508–1.213)	0.275		
	Unknown	1 (reference)			
Lymphovascular invasion (LVI)	−	1 (reference)		1 (reference)	
	+	3.558 (2.275–5.567)	0.001 *	1.596 (0.732–3.479)	0.240
Primary tumor localization	Uterine corpus	1 (reference)		1 (reference)	
	Lower Uterine Segment	2.438 (1.565–3.797)	0.001 *	1.799 (0.881–3.672)	0.107
Metastasis	−	1 (reference)		1 (reference)	
	+	7.555 (4.043–14.119)	0.001 *	4.620 (2.128–10.030)	0.001 *
Surgery	−	1 (reference)		1 (reference)	
	+	0.117 (0.061–0.224)	0.001 *	0.198 (0.063–0.621)	0.005 *
Pelvic Radiotherapy	−	1 (reference)			
	+	0.989 (0.637–1.535)	0.962		
Brachytherapy	−	1 (reference)			
	+	0.589 (0.295–1.777)	0.134		
Adjuvant Therapy	−	1 (reference)			
	+	1.036 (0.672–1.598)	0.873		
Age at Diagnosis		1.082 (1.055–1.110)	0.001 *		
Body Mass Index (BMI)		1.002 (0.963–1.042)	0.932		

(−): negative, (+): positive. * *p* < 0.05.

## Data Availability

The data underlying this article will be shared at reasonable request to the corresponding author.

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
