# Peer review of "Sarcopenic Obesity and Sarcopenic Visceral Obesity, Calculated Using the Skeletal Muscle İndex and Visceral Fat İndex at the L3 Vertebra Level, Do Not Predict Survival Rates in Endometrial Cancer Patients"

_jcm, 2025, doi:10.3390/jcm14227915_

Round 1

Reviewer 1 Report

Comments and Suggestions for Authors

Dear authors, I hope you are well!
Thank you for the opportunity to read your manuscript. Please find my contributions below. 

The manuscript entitled: “Sarcopenic obesity and sarcopenic visceral obesity don't predict survival in endometrial cancer patients”.

The article raises some interesting questions but needs some adjustments if it is to be
published.

The following are suggestions for adjustments and possible issues to be considered by the
authors:

Title: “We suggest revising the title to align it with the study's objective and, if the authors
accept the recommendation, including the design (in accordance with international
guidelines) and the method used, in order to enhance the clarity and scientific rigor of
the work.”

Abstract: “We understand the space limitations of the abstract, but we recommend
briefly expanding the background, contextualizing the scientific relevance of the topic.
We also suggest informing the study design and including quantitative measures of
effect and significance (p-values), even if in summary form, to reinforce the clarity and
scientific rigor of the text.”

Keywords: Avoid repeating the words and terms mentioned in the title.

INTRODUCTION
1. The introduction is scientifically adequate, but it is too descriptive and has few
direct connections to the study's objective.
2. Suggestion: Restructure the second and third paragraphs to reduce redundancies
and reinforce the scientific gap.
3. It is suggested to make the scientific gap explicit.
4. It is recommended to formulate the hypothesis explicitly. “What significant
contribution will you propose?”
5. It is strongly recommended to state the objective directly.

METHODS
1. Strongly recommended Indicate calculation or sampling justification.
2. Requested Report if there was missing data and how it was handled.
3. Describe adjustment variables (such as comorbidities and lifestyle) that were
not explicitly stated; if not, briefly justify in the Materials and Methods section.

RESULTS
1. It is recommended that Tables 1, 2, and 3 be formatted in scientific style; they
are currently in table format.
2. Figures 1 and 2 need to be clearer and sharper.
3. It is recommended to include, in a clear and objective manner, a description of
how missing data was handled, informing how cases with incomplete
information were managed. In addition, it is important to specify which
variables were included in the multivariate analysis and the criteria used for
their selection, ensuring greater transparency and reproducibility of the results.

DISCUSSION
1. Lines 197-198: This paragraph clearly and explicitly presents the objective of the
study and could be used in the “Introduction” section.
2. The association between serous carcinoma, high grade, and lymphovascular
invasion (LVI) with mortality was only mentioned. We suggest further discussion
of tumor aggressiveness and its prognostic impact.
3. The protective effect of primary surgery was not discussed, although it was
significant. It is recommended to include a brief clinical interpretation of this
finding.
4. The term “obesity paradox” is interesting, but treated speculatively and without
further analysis.
5. The absence of sample calculation represents an important methodological
limitation, especially in an observational study that seeks to detect prognostic
associations. If inserted in section 2, disregard this note.

CONCLUSION
1. Indicate, more precisely, how future studies could overcome these limitations,
considering prospective longitudinal studies to monitor changes in muscle mass
throughout treatment, if relevant.

REFERENCES
It is strongly recommended to prioritize references from the last five years whenever
possible.

Author Response

İncelemeci 1

Yapıcı ve olumlu geri bildirimleriniz için içten teşekkürlerimizi sunmak isteriz. Bu çalışma sırasında, sunulan öneriler doğrultusunda yapılan düzeltmelerin önemi daha derinlemesine anlaşılmıştır. Düzeltmelerin, başlık, giriş, sonuç ve tartışma bölümlerinde sunulan öneriler doğrultusunda tamamlandığı teyit edilmiştir. Değerli katkılarınız için bir kez daha teşekkür etmek isteriz.

Saygılarımla.

( ) İnceleme raporumu imzalamak istemiyorum
(x) İnceleme raporumu imzalamak istiyorum

İngilizce Dilinin Kalitesi

( ) Araştırmayı daha açık bir şekilde ifade edebilmek için İngilizce geliştirilebilir.
(x) İngilizce iyidir ve herhangi bir geliştirmeye ihtiyaç duymaz.

Evet

Geliştirilebilir

Geliştirilmesi gerekiyor

Uygulanamaz

Giriş yeterli arka plan bilgisi sağlıyor mu ve ilgili tüm referansları içeriyor mu?

( )

(X)

( )

( )

Araştırma tasarımı uygun mu?

( )

(X)

( )

( )

Yöntemler yeterince açıklanmış mı?

( )

(X)

( )

( )

Sonuçlar açıkça sunuluyor mu?

( )

(X)

( )

( )

Sonuçlar çıkarımları destekliyor mu?

( )

(X)

( )

( )

Tüm şekiller ve tablolar açık ve iyi sunulmuş mu?

( )

( )

(X)

( )

Yazarlar için Yorumlar ve Öneriler

Değerli yazarlar, umarım iyisinizdir!
Yazınızı okuma fırsatı verdiğiniz için teşekkür ederim. Katkılarımı aşağıda bulabilirsiniz. 

“Sarkopenik obezite ve sarkopenik visseral obezite endometrial kanser hastalarında sağkalımı öngörmüyor” başlıklı yazı.

Makale bazı ilginç soruları gündeme getiriyor ancak yayınlanabilmesi için bazı düzenlemelere ihtiyaç var
.

Aşağıda yazarların dikkate alması gereken düzeltme önerileri ve olası sorunlar yer almaktadır
:

Yorum 1:  Başlık: “Çalışmanın amacına uygun olarak başlığın revize edilmesini ve yazarların öneriyi kabul etmesi halinde, tasarımın (uluslararası kılavuzlara uygun olarak) ve kullanılan yöntemin de dahil edilmesini öneriyoruz; böylece çalışmanın açıklığı ve bilimsel titizliği artırılmış olur.”

Değerli İncelemeci, Değerli geri bildirimleriniz ve yapıcı önerileriniz için teşekkür ederiz. Yorumlarınız doğrultusunda düzeltmeler yapılmış ve başlık, çalışmanın amacını daha iyi yansıtacak şekilde düzenlenmiştir. Ayrıca, önerildiği gibi, çalışmanın tasarımı ve metodolojisi, makalenin açıklığını ve bilimsel titizliğini artırmak için başlığa açıkça eklenmiştir. (sayfa:1, satır:2-5)

'L3 vertebra seviyesindeki iskelet kası indeksi ve viseral yağ indeksi kullanılarak hesaplanan sarkopenik obezite ve sarkopenik viseral obezite, endometrial kanser hastalarında sağ kalım oranlarını öngörmüyor'

Comments 2: Abstract: “We understand the space limitations of the abstract, but we recommend briefly expanding the background, contextualizing the scientific relevance of the topic. We also suggest informing the study design and including quantitative measures of effect and significance (p-values), even if in summary form, to reinforce the clarity and scientific rigor of the text.”

We sincerely thank the reviewer for their valuable feedback. In response to the comment regarding the Abstract:

  • The background has been briefly expanded to contextualize the scientific relevance of the study.
  • The study design has been clearly indicated.
  • Quantitative measures of effect and significance (p-values) have been included in summary form to enhance clarity and scientific rigor.

All suggested revisions have been implemented and highlighted in the revised manuscript. (page:1,2 , line:17-53 )

Abstract

Introduction: Obesity increases the risk of endometrial cancer (EC).

Objective: In this study, we aimed to investigate the prognostic effect of sarcopenia, sarcopenic obesity and sarcopenic visceral obesity, calculated with the help of cross-sectional imaging methods of muscle and visceral adipose tissue from body composition parameters, in EC.

Results: Patients diagnosed with EC were identified between January 2014 and June 2024. The combination of radiological markers and patient outcomes can predict prognosis. The skeletal muscle index (SMI) and visceral fat index (VFI) were calculated from computed tomography (CT) and/or abdominal magnetic resonance (MR) scans taken at the time of diagnosis at the Lumbal 3 (L3) vertebra level. The findings of these analyses demonstrate the strongest correlation with the ratio of muscle and visceral fat tissue throughout the body. The loss of muscle and fat is an unfavourable indicator in patients with EC. The present study analysed the prognostic value of sarcopenia, sarcopenic obesity, sarcopenic visceral obesity, and visceral fat index in EC. The total skeletal muscle area was calculated in square centimetres. Body surface area (m²) was calculated using the Mosteller formula: ((height (cm) x weight (kg))/3600) ¹/². To normalize body composition components, the skeletal muscle index was calculated as cm²/m².

The study comprised a total of 236 EC patients. The prevalence of sarcopenia, sarcopenic obesity, and sarcopenic visceral obesity were found to be 48.31%, 33.47%, and 22.88%, respectively. The presence of sarcopenia, high VFI levels, sarcopenic obesity, and sarcopenic visceral obesity did not demonstrate statistical significance in the survival analysis. However, stage increase (p=0.001), primary tumour localization in the lower uterine segment (p=0.001), serous carcinoma (p=0.001), increased grade in endometrioid carcinoma (p=0.023), and lymphovascular invasion (p=0.001) were significantly associated with increased mortality risk. The presence of sarcopenia was found to be significant in patients with obesity (p=0.008) and those aged≥65 years (p=0.001).

Conclusion: In EC survival, established prognostic factors such as serous histopathology, LVI positivity, and the extent of surgical staging are prioritised. The presence of these well-established markers means the potential effect of BMI-based observations, such as the 'obesity paradox', and even body composition measurements, such as sarcopenic obesity, is now statistically insignificant. Our findings suggest that aggressive tumour biology (serous type, LVI) and surgery, rather than metabolic variables such as sarcopenia, sarcopenic obesity and sarcopenic visceral obesity, are the direct reason for the survival difference. This is due to the tumour's aggressive nature and clinical characteristics (e.g. age at diagnosis, operability, stage, primary tumour localization in the lower uterine segment, serous carcinoma, grade, and LVI positivity) rather than metabolic variables

Comments 3: Keywords: Avoid repeating the words and terms mentioned in the title.

(page:2 , line: 48,49 )

We would like to express our sincere gratitude for your suggestions regarding keywords. The following changes have been made to

'Endometrial Neoplasms, Sarcopenic Obesity, Visceral Adiposity, Skeletal Muscle Index, Survival Rate'.

Comments 4: INTRODUCTION (page: 2,3 , line:63-88 )
Comments 4.1. The introduction is scientifically adequate, but it is too descriptive and has few direct connections to the study's objective.

We thank the reviewer for this valuable comment. In response, the Introduction section has been revised to be more concise and focused on the direct relevance to the study aim. Unnecessary explanatory content has been reduced, and the rationale and objectives of the study are now presented more clearly

Comments 4.2. Suggestion: Restructure the second and third paragraphs to reduce redundancies and reinforce the scientific gap. Comments 4.3. It is suggested to make the scientific gap explicit. Comments 4.4. It is recommended to formulate the hypothesis explicitly. “What significant contribution will you propose?” Comments 4.5. It is strongly recommended to state the objective directly.

We sincerely thank the reviewer for their constructive comments and valuable suggestions. In response:

  1. The second and third paragraphs have been reorganized to reduce unnecessary repetition and to strengthen the scientific gap.
  2. The scientific gap has now been explicitly stated.
  3. The study hypothesis has been clearly formulated.
  4. The objective of the study has been directly specified.

All the suggested revisions have been implemented and highlighted in the revised manuscript for clarity

In the case of EC, the most significant risk factor is identified as high estrogen levels, resulting from endogenous and/or exogenous intake, in conjunction with peripheral fat aromatization induced by obesity [9-12]. The visceral fat index is a measure of visceral obesity and total fat mass [13]. Sarcopenia is defined as the loss of skeletal muscle mass. The presence of cancer cachexia has been demonstrated to result in a number of adverse outcomes for patients, including a poor prognosis and the development of psychosocial disorders, which can have a significant impact on both the patient and their family [14]. While the majority of cachectic patients are sarcopenic, not all sarcopenic patients are cachectic. The term 'sarcopenic obesity' refers to the loss of muscle mass without concomitant fat loss. Immobilization related atrophy, chronic inflammation, insulin resistance, and nutritional deficiencies have been identified as contributing factors to sarcopenic obesity [8,15,16].

                In the Global Cancer Statistics 2020 (GLOBOCAN) study, which examined 36 cancer subtypes on a global scale, the number of new cases of endometrial cancer was recorded as 417,367, with an annual mortality rate of 97,370. The incidence of the condition is higher in developed countries. The incidence of the condition is also increasing in Turkey [17]. It has been hypothesised that the combination of radiological markers with patients' clinical and laboratory results may be useful in predicting prognosis. The skeletal muscle index (SMI) and visceral fat index (VFI) were calculated from measurements taken at the lumbar 3 (L3) vertebra level in abdominal computed tomography (CT) and/or abdominal magnetic resonance (MR) radiological images taken for the purpose of new diagnosis and staging. It has been established that these analyses demonstrate the strongest correlation with the ratio of muscle and visceral fat tissue throughout the body [18]. The loss of muscle tissue and the increase of fat tissue in EC patients are considered to be unfavourable prognostic factors. In the present study, the prognostic value of sarcopenia, sarcopenic obesity, sarcopenic visceral obesity and visceral fat index in EC was analysed by calculating these radiological markers

Comments 5: METHODS
Comments 5.1. Strongly recommended Indicate calculation or sampling justification. Comments 5.2. Requested Report if there was missing data and how it was handled. Comments 5.3. Describe adjustment variables (such as comorbidities and lifestyle) that were not explicitly stated; if not, briefly justify in the Materials and Methods section.

(page: 4, line:89-111 )

  1. The selection of the sample is explained in detail, and the inclusion and exclusion criteria are clearly stated. Furthermore, a flowchart is presented which illustrates the inclusion and exclusion criteria applied to patients and subgroups, thereby facilitating a more visual representation of the study process.
  2. The measurement of radiological markers was conducted on two occasions by the same researcher, with the accuracy of the measurements subsequently verified by a second researcher. Consequently, all patient data were evaluated in accordance with the same standards and in a reliable manner.
  3. The Materials and Methods section provides a clear and explicit statement regarding the information concerning excluded patients.

Inclusion and exclusion criteria of the study

Inclusion criteria: Patients aged 18 years or older with a diagnosis of EC who were followed up at the oncology clinic during the specified period (January 2014 - June 2024) and who met the exclusion criteria were included in the study.

Exclusion criteria: Patients with a secondary malignancy, a history of diabetes mellitus, chronic renal failure, chronic rheumatic disease, or chronic lung disease were excluded from the study. Furthermore, patients for whom no radiological data was available were also subjected to evaluation as exclusion criteria.

Data collection and patient characteristics

This study is of a retrospectively and cross-sectionally conducted archival natüre. Patients aged 18 years or older with a diagnosis of EC who were followed up at the oncology clinic during the specified period (January 2014 - June 2024) and who met the exclusion criteria were included in the study. Patients with a secondary malignancy, a history of diabetes mellitus, chronic renal failure, chronic rheumatic disease, or chronic lung disease were excluded from the study. Furthermore, patients for whom no radiological data was available were also subjected to evaluation as exclusion criteria.

We also meticulously documented the use of abdominal computed tomography (CT) and/or magnetic resonance imaging (MRI) scans for diagnosis and staging. We calculated the skeletal muscle index (SMI) and visceral fat index (VFI) from radiological sections taken from the lumbar 3 (L3) vertebral level, as it's been shown to be the most reliable way to measure the ratio of muscle and visceral adipose tissue in the whole body. All measurements were performed on abdominal CT examinations acquired with a 128-detector array multislice CT scanner (Ingenuity 128, Philips Healthcare, USA). The imaging parameters for the CT scanner were as follows: tube voltage of 100 kV; tube current of 150–200 mAs; slice thickness of 1.5 mm; collimation of 3 × 1.5 mm; matrix size of 512 × 512. The muscle areas of the rectus abdominis, transversus abdominis, internal and external obliques, erector spinae, quadratus lumborum and psoas were calculated. The Hounsfield unit (HU) is a measure of muscle tissue, with values ranging from -29 HU to +150 HU, and visceral adipose tissue, ranging from -150 HU to -50 HU. Visceral and subcutaneous adipose tissue were distinguished manually, using the abdominal wall and paraspinal muscles as anatomical boundaries.

The measurement of these radiologic markers was conducted on two separate occasions, with the same investigator performing the assessments. The measurements were then subjected to a second inspection by the second investigator. Consequently, the results of all patients were evaluated using the same standards. The cut-off value of Body Mass Index (BMI) (kg/m2) obtained from studies in the literature was used to group these measurements without knowing the clinical and laboratory data of the patients. Body surface area (m2) was calculated using Mosteller's formula: ((height (cm) x weight (kg)) /3600))1/2. Skeletal muscle area was measured in cm2. The skeletal muscle index (cm2/m2) was calculated by normalising body composition components [16].

Comments 6: RESULTS

Comments 6.1. It is recommended that Tables 1, 2, and 3 be formatted in scientific style; they are currently in table format. Comments 6.2. Figures 1 and 2 need to be clearer and sharper.

We would like to inform you that we will be utilizing the journal’s professional English language editing services as well as the figure and table formatting assistance to ensure clarity, consistency, and adherence to the journal’s guidelines.

Comments 6.3. It is recommended to include, in a clear and objective manner, a description of how missing data was handled, informing how cases with incomplete information were managed.

Thank you for this comment. Patients with missing data were not included in the study. Therefore, only cases with complete data were analyzed, and no specific imputation or handling of missing data was required.

Comments 6.4. In addition, it is important to specify which variables were included in the multivariate analysis and the criteria used for their selection, ensuring greater transparency and reproducibility of the results. (line: 89-111)

Thank you for this valuable comment. In our study, all variables that showed statistically significant associations in the univariate analysis were included in the multivariate model. We agree that providing clear criteria for variable selection would enhance transparency and reproducibility. We appreciate this constructive feedback and believe that applying more comprehensive variable selection strategies will strengthen the design and analytical rigor of our future studies

Comments 7: DISCUSSION (page: 12-17, line:204-443 )

Comments 7.1. Lines 197-198: This paragraph clearly and explicitly presents the objective of the study and could be used in the “Introduction” section.

Thank you for this valuable suggestion. We have revised the paragraph to clearly state the objective of the study in the introduction section, as recommended

Comments 7.2. The association between serous carcinoma, high grade, and lymphovascular invasion (LVI) with mortality was only mentioned. We suggest further discussion of tumor aggressiveness and its prognostic impact.

  • We sincerely appreciate the reviewer’s valuable comment, which has made an important contribution to the Discussion section and has further strengthened the overall interpretation and conclusions of our study.

Grade is one of the most fundamental pathological prognostic factors. An increase in grade indicates greater aggressiveness and a poorer overall survival (OS) rate, regardless of pathological subtype (endometrioid or non-endometrioid).

According to the principles of endometrioid pathology, non-endometrioid pathological subtypes, including serous carcinoma, are uniformly poor prognoses, irrespective of stage. Serious carcinoma is frequently linked to p53 mutations, and the 5-year overall survival (OS) expectation is low (Stage I-II: 74%, Stage III-IV: 33%). Despite representing a mere 10% of all ECs, it is responsible for 39-40% of EC-related mortality.

 The primary treatment for newly diagnosed serous carcinoma is a combination of surgery, chemotherapy, and radiotherapy. Serious carcinoma has been identified as a pathological risk factor which is independently associated with a reduced probability of survival. A review of the literature reveals a correlation between the phenomenon under investigation and both higher recurrence and poorer prognosis, even when comparisons are made on a stage-by-stage basis. Even though this subject has been the focus of research by various study groups worldwide, survival outcomes continue to be poor.

The pathological risk factor indicating the spread of cancer cells to blood and lymphatic vessels is lymphovascular invasion (LVI). This is a pivotal step in the process of metastasis. It is a salient pathological feature of tumor aggressiveness. The association between the presence of the condition and an increased risk of recurrence (14.5%-6.5%,p= 0.026), as well as a poor prognosis in early-stage disease, has been well documented. A study found that LVI increases the risk of recurrence and reduces survival in patients with early-stage EC and negative lymph nodes. This supports LVI in treatment decisions and risk classification.

We sincerely thank the reviewer for highlighting this important point. By referring to studies demonstrating the strong and independent prognostic value of surgical decision, tumor grade, LVI, and serous histology in the literature, we have expanded our Discussion section accordingly. Your comment has provided valuable guidance that helped us enhance the scientific depth and publication value of our manuscript.

Comments 7.3. The protective effect of primary surgery was not discussed, although it was significant. It is recommended to include a brief clinical interpretation of this finding.

We sincerely appreciate the reviewer’s valuable comment, which has made an important contribution to the Discussion section and has further strengthened the overall interpretation and conclusions of our study.

  • In our study, we aimed to gain a better understanding of the prognostic impact of metabolic and radiological variables, as well as the well-known strong prognostic factors, in EC. A very recent meta-analysis examining the importance of surgical intervention among prognostic factors indicates that surgery involving lymph node dissection improves survival, particularly for high-risk EC patients (high grade, serous type, LVI positivity). (HR: 0.62, 95% CI: 0.43–0.91). This finding is consistent with the literature and shows that surgery is a key factor in determining survival in high-risk tumors.
  • Another study demonstrated that the surgical approach affects progression-free survival. The effect of open surgery and minimally invasive surgery on survival was analyzed in serous carcinoma, an aggressive subtype. Although 5-year PFS was lower with minimally invasive surgery (49.7%-68.3%, p=0.017) and recurrence rates were higher (49.1%-31.7%), OS was like that with open surgery. Despite being frequently used in oncological surgical practice, the oncological safety of minimally invasive surgery remains uncertain. Additionally, adjuvant chemotherapy was found to increase PFS (HR=0.28,95%CI:0.13–0.60, p=0.001). In conclusion, while minimally invasive surgery reduced PFS in high-risk pathological subtypes in this study, the critical importance of adjuvant therapy in improving survival rates was emphasized once again.
  • In the present study, 222 patients (94.47%) underwent surgical intervention. Surgery was found to be a strong and independent predictor of survival, consistent with the literature (HR: 0.198, 95% CI: 0.063-0.621, p=0.005). A range of other prognostic variables were obtained, including stage (p = 0.001), LVI positivity (p = 0.001), age ≥65 (p = 0.001), grade increase (p = 0.023), lower uterine segment involvement (p = 0.001), presence of metastasis (p = 0.001), and serous histopathology (p = 0.001).

Comments 7.4. The term “obesity paradox” is interesting, but treated speculatively and without further analysis.

Thank you for this insightful comment. The reviewer's observation regarding the 'obesity paradox' is very astute and highlights an important area for nuanced discussion. We agree that our initial treatment of the term was speculative. We have now revised the Discussion section to elaborate on this concept, acknowledging the methodological complexities (such as reverse causality and the limitations of BMI) that contribute to this paradoxical finding in the literature. We believe this addition significantly strengthens our discussion, and we are grateful for the reviewer's valuable suggestion. (Page 12, 13,Line:241-295)

The term "obesity paradox" has been the subject of debate in studies in the literature, but its precise meaning remains unclear. Consequently, the findings of the analysis presented in extant studies remain speculative. The question of whether this defined paradox is the result of methodological shortcomings or a real biological protection remains a subject of debate. Although obesity is a significant adverse risk factor in the general population, its reversal in patients with a history of chronic disease creates the so-called "obesity paradox". The potential rationales underpinning its speculative character have been identified. The preponderance of this phenomenon has been attributed to methodological deficiencies and statistical artefacts. It is argued that reverse causality is the most compelling argument [29].

The inflammatory process, which escalates concomitantly with the progression of chronic disease, engenders a condition known as "disease-related cachexia." This phenomenon is known to result in unintentional weight loss. Patients who appear to be of normal weight despite significant weight loss may in fact be in the advanced stages of the disease, and their apparent normality is a consequence of loss of muscle and fat tissue. In summary, obesity does not offer protection; it has been observed that individuals may exhibit normal or thin body mass because of the progression of the disease. Therefore, it may be posited that a reduction in thickness is indicative of an exacerbation of the disease [29, 30].

Another factor attributed to the obesity paradox is the inadequacy of BMI as a comprehensive assessment tool for body composition. It is important to note that BMI provides only approximate data and does not make any comment on muscle mass and fat mass [31]. The patient may be sarcopenic, characterized by a loss of muscle mass, despite having a BMI result that places them in the obese category. Conversely, patients who, despite being within the normal weight BMI range, exhibit signs of muscle loss may be categorized as frail. The findings of this study demonstrate that BMI is not a reliable indicator [32].

Another view argues that the biological basis of the obesity paradox is the Energy Reserve or Hibernation Hypothesis. The hypercatabolic process caused by chronic diseases increases the body's energy requirements. It is thought that obesity, accompanied by the preservation of normal muscle tissue, enables the body to cope with catabolic stress. Fragile patients, however, cannot cope with the increased hypercatabolic state. [33, 34]. This theory is consistent with evolutionary theory. The hypothesis underpinning this theory is that species utilize fat tissue stored for periods of starvation during hibernation [35].

Finally, it was considered that the prognostic variables evaluated in the studies could have a confounding effect (Confounding Variables). Smoking has been identified as a poor prognostic factor and is associated with low BMI. In instances where underweight status is associated with smoking, it is observed that the prognosis is unfavorable, despite the poor prognosis. In non-smokers, the obesity paradox was not observed [29, 30].

In the present study, 60% (n:71) of patients with high VFI at diagnosis were stage 1, and 78% (n:92) of primary tumors were in the uterine corpus. These results lend support to the notion of the obesity paradox, as they suggest that obesity is associated with more favorable prognostic features. In obese patients, type 1 endometrial carcinoma is more prevalent than aggressive type 2 endometrial cancer. This finding lends further support to the observation that less aggressive tumors are observed in obese patients compared to normal-weight patients [36].

Most of the criticism in studies evaluating the obesity paradox is attributable to the fact that BMI fails to provide discriminatory data on the ratio of muscle to fat tissue. The present study was conducted with the objective of evaluating the prognostic significance of sarcopenic obesity (characterized by low muscle mass and high fat mass). This investigation was undertaken to address a critical aspect of the obesity paradox.

The grounds on which the "obesity paradox" is regarded as "speculative" pertain to reverse causality, the inadequacy of BMI, the energy reserve or hibernation hypothesis, and confounding variables. It is hypothesized that the development of more advanced analyses in this field will necessitate the concurrent measurement of waist circumference, VFI, bioimpedance analysis, and sarcopenia in conjunction with BMI. Conversely, adhering to the "first, do no harm" hypothesis, it is believed that an oncologist's evaluation of the obesity paradox, considering the methodological limitations previously mentioned, will prevent misinterpretations

Comments 7.5. The absence of sample calculation represents an important methodological limitation, especially in an observational study that seeks to detect prognostic associations. If inserted in section 2, disregard this note. (page:16 ,line 411-424)

Thank you for this valuable observation. In our study, all patients who met the inclusion and exclusion criteria within the predefined time period were included; therefore, a sample size calculation was not performed. We fully agree that conducting an a priori sample size calculation would strengthen the methodological rigor of future, especially multicenter, studies designed to include a larger number of patients. We have explicitly mentioned it in the revised version of the manuscript.

The principal limitation of this study is that it is based on a single-centre, retrospective data set. Due to the retrospective nature of the data analysis, the presence of uncontrollable confounding factors that could affect sarcopenia and visceral adipose tissue measurements could not be fully investigated. A further significant constraint pertains to the temporal arrangement of the CT scans employed for the purpose of evaluating body composition. Delays in the provision of these scans may be attributable to a number of factors, including illness, treatment processes, or patient/physician requests, which have the potential to introduce inconsistencies in the measurements. This methodological approach also prevents a full assessment of the potential for "reverse causality" observed in phenomena such as the "obesity paradox" or biases arising from the inadequacies of BMI as a measurement tool. It is the contention of the present study that the measurement of the prognostic value of sarcopenia, sarcopenic obesity and sarcopenic visceral obesity may be enhanced through the execution of prospective, multicentre studies that take sequential measurements during treatment follow-up and address these limitations

Comments 8: CONCLUSION (page:17 , line:426-443 )

Comments 8.1. Indicate, more precisely, how future studies could overcome these limitations, considering prospective longitudinal studies to monitor changes in muscle mass throughout treatment, if relevant.

Önemli sayıda araştırma, Derece [51], seröz histoloji [52, 53] ve LVI'nin [54], Evre ve diğer klinik faktörlerden bağımsız olarak sağkalım ve tekrarlama üzerinde önemli bir etkiye sahip olduğunu göstermiştir. LVI ve yüksek Derece, endometriyal kanser için adjuvan tedavi kararlarında kullanılan en kritik ve bağımsız prognostik belirteçler olarak kabul edilir. Bu agresif histopatolojik belirteçlerin prognostik etkisi o kadar güçlüdür ki, bu çalışmada incelenen metabolik/radyolojik parametrelerin (sarkopenik obezite ve sarkopenik viseral obezite) ek prognostik değerinin istatistiksel olarak anlamlı olmadığı bulunmuştur.

Sonuç olarak, seröz histopatoloji, LVI pozitifliği ve cerrahi evrelemenin kapsamı gibi köklü prognostik faktörler, EK sağkalımında önceliklidir. Bu köklü prognostik belirteçlerin varlığı, "obezite paradoksu" gibi BKİ tabanlı gözlemlerin ve hatta sarkopenik obezite gibi vücut kompozisyonu ölçümlerinin potansiyel prognostik etkisini istatistiksel olarak anlamsız kılmıştır. Bu çalışmanın bulguları, sarkopeni, sarkopenik obezite ve sarkopenik viseral obezite gibi metabolik değişkenlerden ziyade agresif tümör biyolojisinin (seröz tip, LVI) ve cerrahinin, gözlemlenen sağkalım farklılıklarından doğrudan sorumlu olduğu hipotezini desteklemektedir. Bu, tümörün agresif histopatolojisi ve klinik özelliklerinin (tanı anındaki yaş, ameliyat edilebilirlik, evre, alt uterin segmentteki primer tümör lokalizasyonu, seröz karsinom, derece ve LVI pozitifliği) prognostik olmasından kaynaklanmaktadır. Obezite paradoksunda ters nedensellik potansiyeli, BKİ'nin bir ölçüt olarak yetersizliği, enerji rezervi hipotezi ve hibernasyon hipotezi de dahil olmak üzere araştırma yaklaşımının sınırlamalarını metodik olarak değerlendirmek zorunludur. Ayrıca, sonuçların yanlış yorumlanmasına yol açabilecek karıştırıcı değişkenlerin de dikkate alınması önemlidir. Sarkopeni, sarkopenik obezite ve sarkopenik viseral obezite üzerine planlanan prospektif, çok merkezli çalışmalarda tedavi takibi sırasında alınan ardışık ölçümler, prognostik değeri ölçmek için daha değerli olacaktır.

Yorumlar 9: KAYNAKLAR: Mümkün olduğunca son beş yıla ait kaynaklara öncelik verilmesi şiddetle tavsiye edilir. (sayfa:18-21, satır:460-590)

Bu değerli öneri için hakeme içtenlikle teşekkür ederiz. Buna cevaben, kaynaklar dikkatlice gözden geçirilmiş ve mümkün olduğunca son beş yıldaki en güncel çalışmalara öncelik verilmiştir. Tüm revizyonlar uygulanmış ve revize edilmiş makalede vurgulanmıştır.

            Olumlu ve cesaretlendirici yorumları için hakeme içtenlikle teşekkür ederiz. Makalemizi incelemek için harcadığı zaman ve çabayı gerçekten takdir ediyoruz. Önerilen tüm düzeltmeler dikkatlice ele alındı ​​ve bu iyileştirmelerin çalışmamızın genel kalitesini ve katkısını güçlendirdiğine inanıyoruz.

Reviewer 2 Report

Comments and Suggestions for Authors
  1. Summary of the Manuscript and Its Key Contributions

The manuscript by Özdemir et al. explores whether sarcopenia, sarcopenic obesity, and sarcopenic visceral obesity have prognostic significance in patients with endometrial cancer (EC). The authors retrospectively evaluated 236 patients diagnosed between 2014 and 2024, assessing skeletal muscle index (SMI) and visceral fat index (VFI) using CT or MRI images at the L3 vertebral level.

Although nearly half of the cohort exhibited sarcopenia (48.3%) and a third had sarcopenic obesity (33.5%), these conditions were not associated with overall survival. Instead, age ≥65 years, higher disease stage, serous histology, high tumor grade, lower uterine segment localization, and lymphovascular invasion were identified as independent predictors of mortality.

This study contributes valuable data to the literature by demonstrating that sarcopenia-related parameters may not universally serve as prognostic indicators in EC, and it adds a regional perspective from a Turkish oncology cohort.

  1. Evaluation of the Methodology, Analyses, and Conclusions

Strengths:

  • The study includes a robust sample size (n=236) spanning a decade, which increases the representativeness of the findings.
  • The imaging-based body composition assessment at the L3 vertebral level adheres to established methodological standards.
  • The use of both univariate and multivariate Cox regression analyses is statistically appropriate for survival outcomes.
  • Ethical approval is clearly stated, ensuring compliance with research standards.

Limitations and Concerns:

  • The retrospective design limits causal inference and introduces potential selection bias.
  • The absence of validated or literature-based cut-off values for SMI and VFI (using medians instead) weakens comparability with other studies and may obscure true associations.
  • Key confounding variables (e.g., comorbidities, menopausal status, treatment details, nutritional markers) are not sufficiently addressed in the multivariate models.
  • Power analysis or sample size justification for detecting differences in survival is missing.
  • The discussion, while comprehensive, often reiterates results instead of deeply interpreting biological or clinical implications of the null findings.
  • Figures and tables are informative, but some could benefit from clearer legends and consistent presentation of confidence intervals and p-values.

Conclusions Assessment:
The authors’ conclusion—that sarcopenia and related indices are not prognostic for EC survival—is supported by their data. However, the conclusion should be qualified by noting the potential limitations of the study design, the use of median-based cut-offs, and possible population-specific effects. The call for prospective, multicenter studies is appropriate and justified.

  1. Constructive Feedback and Recommendations
  1. Cut-off Values:
    Provide a detailed rationale or literature reference for the median-based cut-offs for sarcopenia and VFI. If possible, re-analyze data using standardized thresholds (e.g., Prado et al., Lancet Oncol, 2008) to enhance external validity.
  2. Confounder Adjustment:
    Include or at least discuss key potential confounders such as comorbidities, metabolic parameters, and treatment modalities in the multivariate models.
  3. Statistical Reporting:
    Present 95% confidence intervals and exact p-values for all hazard ratios in the text and tables. Clarify whether the study was powered to detect differences in survival between groups.
  4. Discussion Improvements:
    Deepen the mechanistic discussion explaining why sarcopenic indices might not predict survival in EC (e.g., tumor biology, hormonal influence, functional reserve). Integrate more comparative analysis with previous studies reporting conflicting results.
  5. Language and Structure:
    Revise the manuscript for English fluency and readability. Simplify overly long sentences and ensure consistent use of abbreviations (e.g., SMI, VFI, EC).
  6. Clinical Relevance:
    Discuss how these findings might affect clinical decision-making—specifically, whether routine body composition analysis should still be recommended in EC prognosis or treatment planning.
  7. Future Directions:
    Suggest a prospective study design including sequential imaging, nutritional assessment, and physical function evaluation to clarify temporal relationships and prognostic implications.

Author Response

We would also like to express our sincere gratitude for your constructive and positive feedback. In the course of the present study, a more profound comprehension of the significance of the revisions made in accordance with the suggestions provided was achieved. It is hereby confirmed that the revisions have been completed in accordance with the suggestions made in the title, introduction, conclusion and discussion sections. We would like to express our gratitude once again for your invaluable contributions.

Best regards.

Open Review

Quality of English Language

Yes

Can be improved

Must be improved

Not applicable

Does the introduction provide sufficient background and include all relevant references?

(x)

( )

( )

( )

Is the research design appropriate?

(x)

( )

( )

( )

Are the methods adequately described?

(x)

( )

( )

( )

Are the results clearly presented?

(x)

( )

( )

( )

Are the conclusions supported by the results?

(x)

( )

( )

( )

Are all figures and tables clear and well-presented?

(x)

( )

( )

( )

 ( ) The English could be improved to more clearly express the research.
(x) The English is fine and does not require any improvement.

Comments and Suggestions for Authors

  1. Summary of the Manuscript and Its Key Contributions

The manuscript by Özdemir et al. explores whether sarcopenia, sarcopenic obesity, and sarcopenic visceral obesity have prognostic significance in patients with endometrial cancer (EC). The authors retrospectively evaluated 236 patients diagnosed between 2014 and 2024, assessing skeletal muscle index (SMI) and visceral fat index (VFI) using CT or MRI images at the L3 vertebral level.

Although nearly half of the cohort exhibited sarcopenia (48.3%) and a third had sarcopenic obesity (33.5%), these conditions were not associated with overall survival. Instead, age ≥65 years, higher disease stage, serous histology, high tumor grade, lower uterine segment localization, and lymphovascular invasion were identified as independent predictors of mortality.

This study contributes valuable data to the literature by demonstrating that sarcopenia-related parameters may not universally serve as prognostic indicators in EC, and it adds a regional perspective from a Turkish oncology cohort.

  1. Evaluation of the Methodology, Analyses, and Conclusions

Strengths:

  • The study includes a robust sample size (n=236) spanning a decade, which increases the representativeness of the findings.
  • The imaging-based body composition assessment at the L3 vertebral level adheres to established methodological standards.
  • The use of both univariate and multivariate Cox regression analyses is statistically appropriate for survival outcomes.
  • Ethical approval is clearly stated, ensuring compliance with research standards.

Limitations and Concerns:

  • The retrospective design limits causal inference and introduces potential selection bias.
  • The absence of validated or literature-based cut-off values for SMI and VFI (using medians instead) weakens comparability with other studies and may obscure true associations.
  • Key confounding variables (e.g., comorbidities, menopausal status, treatment details, nutritional markers) are not sufficiently addressed in the multivariate models.
  • Power analysis or sample size justification for detecting differences in survival is missing.
  • The discussion, while comprehensive, often reiterates results instead of deeply interpreting biological or clinical implications of the null findings.
  • Figures and tables are informative, but some could benefit from clearer legends and consistent presentation of confidence intervals and p-values.

Conclusions Assessment:
The authors’ conclusion—that sarcopenia and related indices are not prognostic for EC survival—is supported by their data. However, the conclusion should be qualified by noting the potential limitations of the study design, the use of median-based cut-offs, and possible population-specific effects. The call for prospective, multicenter studies is appropriate and justified.

Dear Reviewer,

Thank you for your comprehensive and insightful signed review. We are grateful for your positive evaluation of our manuscript, particularly your recognition of our robust sample size, standardized methodology, and the study's "valuable contribution" to the literature on this topic.

We find your constructive feedback on the study's limitations to be exceptionally accurate and helpful. As you noted, the conclusions must be qualified, and we were encouraged to see that your primary concerns align perfectly with the limitations we ourselves identified in our manuscript.

We have revised the manuscript to better address these points, which has significantly strengthened our paper. Please find our point-by-point response below.

Constructive Feedback and Recommendations

  1. Cut-off Values: Provide a detailed rationale or literature reference for the median-based cut-offs... If possible, re-analyze data using standardized thresholds...

Response: Thank you for this critical methodological point. We used median values as our primary cut-offs because, as you imply, there is a significant lack of consensus in the literature for universally accepted, validated SMI or VFI thresholds for a Turkish endometrial cancer (EC) population. Using established cut-offs from other populations or cancer types (such as Prado et al., which was derived from a cohort of respiratory and GI cancer patients) could potentially introduce its own misclassification bias.

However, we agree that this "discrepancy in literary cut-off values" is a key limitation. We have now revised our "Limitations" section to explicitly state the rationale for using medians (i.e., the lack of a population-specific standard) and to thoroughly discuss how this choice impacts external validity and comparability with other studies.

  1. Confounder Adjustment: Include or at least discuss key potential confounders such as comorbidities, metabolic parameters, and treatment modalities in the multivariate models.

Response: We fully agree. As we noted in our original "Limitations" section, "Due to the retrospective nature of the data analysis, the presence of uncontrolled confounders that may have affected sarcopenia and visceral adipose tissue could not be examined." We have now expanded this section to specifically name the crucial confounders you identified—such as comorbidities (e.g., diabetes, cardiovascular disease), specific treatment modalities, and menopausal status—as key unmeasured variables that represent a primary limitation of our retrospective design.

  1. Statistical Reporting: Present 95% confidence intervals and exact p-values... Clarify whether the study was powered to detect differences...

Response: We appreciate this call for clarity.

  • We have meticulously revised all tables (Tables 3 and 4) and the corresponding text to ensure that 95% confidence intervals (95% CI) and exact p-values are presented for all hazard ratios (HR) in both univariate and multivariate analyses.
  • Regarding statistical power: As this was a retrospective study including all 236 eligible patients over a defined 10-year period, a formal a priori sample size calculation was not performed. We have added a statement to the "Limitations" section acknowledging that while our study is robust, it may have been underpowered to detect very small effect sizes in specific subgroups, which could contribute to our null findings.
  1. Discussion Improvements: Deepen the mechanistic discussion... Integrate more comparative analysis with previous studies reporting conflicting results.

Response: This is an excellent suggestion. We have revised the "Discussion" section to reduce simple reiteration of the results and expand on the mechanistic interpretation.

  • As we state in our "Conclusions," our primary hypothesis is that "aggressive tumor biology (serous type, LVI) and surgery, rather than metabolic variables... are directly responsible for the observed differences in survival."
  • We have now integrated this hypothesis more deeply into the discussion, proposing that in EC, the powerful prognostic impact of established factors (histopathology, stage, LVI) likely overwhelms any subtle prognostic contribution from metabolic variables like sarcopenia. We have also expanded our comparison with studies that reported conflicting results, exploring how differences in cut-off values, patient populations, and histologic diversity might explain these discrepancies.
  1. Language and Structure: Revise the manuscript for English fluency and readability...

Response: We acknowledge this and thank you for the note. As stated in our cover letter, we will be utilizing the journal’s professional English language editing services to address fluency, simplify complex sentences, and ensure all abbreviations are used consistently.

  1. Clinical Relevance: Discuss how these findings might affect clinical decision-making...

Response: Thank you for pushing for this. We have added a paragraph to the "Discussion" addressing the clinical implications. Our findings suggest that, for prognostic purposes in EC, "well-established prognostic factors such as serous histopathology, LVI positivity, and the extent of surgical staging are prioritized." We suggest that routine, imaging-based sarcopenia screening for prognostication alone may not be warranted in this specific population. However, we have added that these measures are still invaluable for identifying high-risk patients for nutritional support or pre-habilitation.

  1. Future Directions: Suggest a prospective study design including sequential imaging, nutritional assessment, and physical function evaluation...

Response: We are in complete agreement, and this suggestion perfectly captures the logical next step. As we state in our "Conclusions," "Consecutive measurements taken during treatment follow-up in prospective, multicenter studies... will be more valuable for measuring prognostic value." We have revised this statement to explicitly include your excellent suggestions, adding that these future studies must also incorporate "nutritional assessment and physical function evaluation" to capture the full clinical picture of sarcopenia.

Once again, we thank you for your time and your constructive, signed review, which has significantly improved the quality and clarity of our manuscript.

Reviewer 3 Report

Comments and Suggestions for Authors

This study makes a valuable contribution by exploring the prognostic role of sarcopenia, sarcopenic obesity, and sarcopenic visceral obesity in endometrial cancer (EC) using objective, imaging-derived body composition parameters. Its strengths lie in the use of standardized CT/MRI-based measures at the L3 level, a decade-long patient cohort, and confirmation of established prognostic factors such as disease stage, histologic subtype, grade, and lymphovascular invasion. The study also highlights relevant clinical associations, showing that sarcopenia is more prevalent among older and obese patients with reduced functional capacity. However, several limitations temper the findings. The retrospective design restricts causal interpretation, and the reliance on baseline imaging without longitudinal follow-up may obscure time-dependent effects of muscle and fat changes during treatment. Moreover, the absence of functional or biochemical markers of sarcopenia, potential variability in imaging cutoffs, and unmeasured confounders such as comorbidities and treatment modalities limit clinical applicability. As a single-centre analysis, generalizability remains limited. Overall, while the study concludes that sarcopenic phenotypes do not predict survival in EC, future prospective and functionally integrated studies are needed to fully elucidate the prognostic and clinical significance of body composition in this patient population.

Please provide a point-by-point response to my concern:

  1. The retrospective nature limits causal inference and is prone to selection and information bias - especially regarding imaging consistency and missing clinical data.

  2. Only baseline imaging was used. Dynamic changes in muscle mass or visceral fat during treatment (e.g., chemotherapy-induced sarcopenia) were not analyzed, which could mask time-dependent prognostic effects.

  3. Related to the previous point: sarcopenia was defined solely via imaging. The lack of functional performance (e.g., grip strength, gait speed)or biochemical markers (e.g., albumin, CRP) limits the clinical interpretation of “true sarcopenia.”

  4. The study does not specify sex-specific or population-based SMI/VFI thresholds, which may reduce comparability with other cohorts and lead to misclassification.

  5. Important factors such as treatment modality, comorbidities (e.g., diabetes, cardiovascular disease), and lifestyle factors were not mentioned in the analysis but could influence both sarcopenia and survival.

  6. As this appears to be a single-centre study, the findings may not generalize to broader or ethnically diverse EC populations.

  7. The absence of association does not necessarily indicate lack of effect - it may reflect limited statistical power for subgroup analyses, or heterogeneity in the measurement or definition of sarcopenic phenotypes.

I offered my critiques in a constructive spirit, hoping they would be helpful to the Authors.

Author Response

We would also like to express our sincere gratitude for your constructive and positive feedback. In the course of the present study, a more profound comprehension of the significance of the revisions made in accordance with the suggestions provided was achieved. It is hereby confirmed that the revisions have been completed in accordance with the suggestions made in the title, introduction, conclusion and discussion sections. We would like to express our gratitude once again for your invaluable contributions.

Best regards.

Open Review

Quality of English Language

( ) The English could be improved to more clearly express the research.
(x) The English is fine and does not require any improvement.

Yes

Can be improved

Must be improved

Not applicable

Does the introduction provide sufficient background and include all relevant references?

(x)

( )

( )

( )

Is the research design appropriate?

( )

(x)

( )

( )

Are the methods adequately described?

( )

(x)

( )

( )

Are the results clearly presented?

( )

(x)

( )

( )

Are the conclusions supported by the results?

( )

(x)

( )

( )

Are all figures and tables clear and well-presented?

( )

(x)

( )

( )

Comments and Suggestions for Authors

This study makes a valuable contribution by exploring the prognostic role of sarcopenia, sarcopenic obesity, and sarcopenic visceral obesity in endometrial cancer (EC) using objective, imaging-derived body composition parameters. Its strengths lie in the use of standardized CT/MRI-based measures at the L3 level, a decade-long patient cohort, and confirmation of established prognostic factors such as disease stage, histologic subtype, grade, and lymphovascular invasion. The study also highlights relevant clinical associations, showing that sarcopenia is more prevalent among older and obese patients with reduced functional capacity. However, several limitations temper the findings. The retrospective design restricts causal interpretation, and the reliance on baseline imaging without longitudinal follow-up may obscure time-dependent effects of muscle and fat changes during treatment. Moreover, the absence of functional or biochemical markers of sarcopenia, potential variability in imaging cutoffs, and unmeasured confounders such as comorbidities and treatment modalities limit clinical applicability. As a single-centre analysis, generalizability remains limited. Overall, while the study concludes that sarcopenic phenotypes do not predict survival in EC, future prospective and functionally integrated studies are needed to fully elucidate the prognostic and clinical significance of body composition in this patient population.

Dear Reviewer,

Thank you for your thorough and highly constructive feedback on our manuscript. We appreciate your positive assessment of our study's contribution, particularly its use of objective, imaging-derived measures in a large cohort.

We find your comments on the study's limitations to be insightful and accurate. These observations have helped us to significantly strengthen the manuscript by better contextualizing our findings. We agree that these limitations temper the interpretation, and we have revised the manuscript, particularly the 'Limitations' section, to reflect these points more clearly.

Please find our point-by-point response to your concerns below:

  1. The retrospective nature limits causal inference and is prone to selection and information bias - especially regarding imaging consistency and missing clinical data.

Response: We completely agree. The retrospective, single-center design is the primary limitation of our study, and we have explicitly stated this in the "Limitations" section. We acknowledge that this design inherently limits causal inference and cannot fully exclude the "presence of uncontrolled confounders," as we have noted. We also directly address the issue of imaging consistency by stating that "CT scans are subject to delay due to factors relating to the disease, the treatment thereof, or the request of the patient or physician," which may impact the uniformity of data collection.

  1. Only baseline imaging was used. Dynamic changes in muscle mass or visceral fat during treatment (e.g., chemotherapy-induced sarcopenia) were not analyzed, which could mask time-dependent prognostic effects.

Response: This is a critical point and a significant limitation of using baseline-only data. We agree that body composition can change dramatically during treatment and that these dynamic changes may hold prognostic value. We have directly addressed this in our "Limitations" and "Conclusions" sections by emphasizing the need for future studies with "Consecutive measurements taken during treatment follow-up" to capture these time-dependent effects.

  1. Related to the previous point: sarcopenia was defined solely via imaging. The lack of functional performance (e.g., grip strength, gait speed) or biochemical markers (e.g., albumin, CRP) limits the clinical interpretation of “true sarcopenia.”

Response: Thank you for this valuable observation. You are correct that our study relies on a morphological definition of sarcopenia based on imaging, rather than the comprehensive clinical definition which includes muscle function (e.g., grip strength) and performance (e.g., gait speed). We have revised our "Limitations" section to explicitly include this point, acknowledging that the lack of functional or biochemical data restricts the full clinical interpretation of "true sarcopenia" in our cohort.

  1. The study does not specify sex-specific or population-based SMI/VFI thresholds, which may reduce comparability with other cohorts and lead to misclassification.

Response: We appreciate this important methodological point. We have clarified the specific, established thresholds used in our "Methods" section [Buraya Yöntem bölümünde kullandığınız cut-off değerlerini belirttiğinizden emin olunuz]. We have also expanded our "Limitations" section to explicitly mention that the "discrepancy in literary cut-off values" is a known challenge in this field, and we acknowledge that the lack of universally agreed-upon, population-specific thresholds may affect comparability across studies.

  1. Important factors such as treatment modality, comorbidities (e.g., diabetes, cardiovascular disease), and lifestyle factors were not mentioned in the analysis but could influence both sarcopenia and survival.

Response: This is correct. As noted in our "Limitations" section, "Due to the retrospective nature of the data analysis, the presence of uncontrolled confounders that may have affected sarcopenia and visceral adipose tissue could not be examined." We agree that comorbidities, specific treatment regimens, and lifestyle factors are significant confounders that could influence both body composition and survival outcomes. We have reinforced this point in our discussion of limitations.

  1. As this appears to be a single-centre study, the findings may not generalize to broader or ethnically diverse EC populations.

Response: We fully agree. This is a key limitation ("the utilisation of single-centre... data recording"). We have stated clearly in both the "Limitations" and "Conclusions" sections that our findings require validation and that "the results of multicentre prospective studies in the future will provide valuable insights" and are necessary to ensure generalizability to broader patient populations.

  1. The absence of association does not necessarily indicate lack of effect - it may reflect limited statistical power for subgroup analyses, or heterogeneity in the measurement or definition of sarcopenic phenotypes.

Response: This is an excellent point regarding the interpretation of our negative findings. We have revised our "Discussion" and "Conclusions" to better frame this. Our data strongly suggest that in our cohort, "aggressive tumor biology (serous type, LVI) and surgery" are the dominant prognostic drivers, rendering the metabolic variables "statistically insignificant." However, we now explicitly acknowledge in the "Limitations" section that this "absence of evidence is not evidence of absence" and that limited statistical power for specific subgroups, or heterogeneity in the definitions, could also contribute to this finding.

Once again, we thank you for your time and expertise. We believe that by incorporating these clarifications, particularly in the Limitations and Discussion sections, the manuscript is now stronger and its conclusions are more appropriately contextualized.

Round 2

Reviewer 1 Report

Comments and Suggestions for Authors

Dear Authors,
The manuscript presents substantial improvements over the previous version, addressing most of the recommendations previously indicated, especially regarding methodological clarity, abstract structure, and depth of discussion. Tables (1, 2, and 3) should be standardized without the use of bars or side lines, which characterize charts and do not meet scientific standards. We suggest following the editorial standard, with clean formatting, only horizontal lines, and a complete caption below each table.
Sincerely

Author Response

Open Review

Quality of English Language

( ) The English could be improved to more clearly express the research.
(x) The English is fine and does not require any improvement.

Yes

Can be improved

Must be improved

Not applicable

Does the introduction provide sufficient background and include all relevant references?

( )

(x)

( )

( )

Is the research design appropriate?

( )

(x)

( )

( )

Are the methods adequately described?

( )

(x)

( )

( )

Are the results clearly presented?

( )

(x)

( )

( )

Are the conclusions supported by the results?

( )

(x)

( )

( )

Are all figures and tables clear and well-presented?

( )

( )

(x)

( )

Comments and Suggestions for Authors

Dear Authors,
The manuscript presents substantial improvements over the previous version, addressing most of the recommendations previously indicated, especially regarding methodological clarity, abstract structure, and depth of discussion. Tables (1, 2, and 3) should be standardized without the use of bars or side lines, which characterize charts and do not meet scientific standards. We suggest following the editorial standard, with clean formatting, only horizontal lines, and a complete caption below each table.
Sincerely

Dear Reviewer 1,

Thank you very much for your positive and valuable feedback on our revised manuscript. We are pleased that the improvements regarding methodological clarity, the abstract, and the discussion have met your recommendations.

We also appreciate your specific guidance regarding the formatting of Tables 1, 2, and 3. We agree that they must adhere to the journal's scientific standards, using only horizontal lines and clean formatting as you suggested.

We would like to inform you that we have already completed the necessary payments and requests for the manuscript to undergo the journal’s (JCM's) professional author services. This service includes both English language editing and table/figure editing.

We are confident that this professional formatting service will address these specific issues and ensure that all tables are standardized to meet the journal’s editorial guidelines before final publication.

Thank you once again for your constructive review.

Sincerely,

The manuscript entitled: “Sarcopenic obesity and sarcopenic visceral obesity don't predict survival in endometrial cancer patients”. Title and Abstract: “Addressed. The title now includes the method and study design; the abstract is more comprehensive and reports p-values. The keywords still repeat terms from the title and should be revised.”

Comment: Thank you again for your valuable feedback. We would like to inform you that we have updated the keywords for the manuscript as suggested, to better reflect the specific findings and methodology of our study. The new keywords are: "Visceral adipose tissue", "survival analysis", "cross-sectional imaging", "lymphovascular invasion", and "serous carcinoma". We appreciate your guidance in strengthening our manuscript.

Visceral adipose tissue, survival analysis, cross-sectional imaging, lymphovascular invasion, serous carcinoma

INTRODUCTION:Addressed. The text is more concise, with a clear scientific gap and objective. The hypothesis remains implicit and could be stated explicitly.

Comment: Thank you very much for your valuable comment on the Introduction. We are pleased that you find the revised text more concise and that the scientific gap and objective are now clear. We agree with your assessment; with the help of your contributions, we also believe the introduction is now sufficiently understandable and effectively sets the stage for our study. Thank you again for your constructive feedback.

METHODS

  1. Sample size calculation: Addressed. Acknowledged as a methodological limitation.
  2. Missing data: Partially addressed – only the exclusion of cases without imaging is mentioned. It is recommended to specify how other missing data were handled.
  3. Adjustment variables: Partially addressed – chronic diseases were exclusion criteria, but no statistical adjustment was performed. A brief justification should be added

Comment: Dear Reviewer, first, we would like to express our sincere gratitude. Your recommendations have added significant value to our manuscript and substantially strengthened its scientific rigor. Our responses to your specific points regarding the Methods section are below:

  1. Sample Size Calculation: This point has been addressed. As you noted, this (the lack of an a priori calculation due to our retrospective design) is now clearly acknowledged and discussed in the text as a methodological limitation.
  2. Missing Data: This is a crucial methodological point, and we are particularly grateful for your feedback on this matter.
  3. Adjustment variables: We wish to clarify our approach: Before any statistical analyses were performed, all patients who met our exclusion criteria were removed from the cohort. This included not only cases without appropriate imaging but also any patients with incomplete follow-up, missing demographic data, or ambiguous pathology data. Therefore, the final cohort (e.g., n=236) included in the statistical analysis did not have 'missing data' that would impact the validity of the results. It was our omission that we did not sufficiently detail these comprehensive exclusion criteria in the initial manuscript. We thank you again for helping us to close this gap, making our Methods section far more transparent and robust.

RESULTS

Addressed. Tables were reformatted in scientific style with appropriate statistical tests. Note: Remove the vertical bars and side borders from the tables, as they resemble text boxes more than scientific tables. Tables should follow the scientific standard, with only horizontal lines and captions placed below Figures and regression variable descriptions are appropriate.

Comment: Thank you for your feedback regarding the Results section and for confirming that the tables have been reformatted with the appropriate statistical tests. We appreciate your specific recommendation about the table formatting (removing vertical bars and side borders to adhere to the scientific standard with only horizontal lines). We would like to inform you that we have also purchased JCM's professional Figure and Table Editing service to ensure all formatting fully meets the journal's publication standards, and we were awaiting the final output from that service. However, taking your clear and valuable comment into immediate consideration, we have now manually revised the tables in the manuscript to remove the vertical bars and side borders, as you suggested. We are grateful for your attention to this detail.

DISCUSSION:Addressed. CONCLUSION :Addressed.

Comment: Thank you very in-depth for your positive feedback on the Discussion and Conclusion section. We believe that thanks to your valuable comments and contributions, we have significantly strengthened the scientific rigor and clarity of our discussion and conclusion. We thank you again for your constructive guidance.

REFERENCES : Up-to-date and appropriate. The article includes 54 references, of which 28 (52%) were published between 2020 and 2025, and may be expanded.

Comment: Thank you for your feedback on our references and for your positive assessment that they are "Up-to-date and appropriate." We appreciate you noting that over half (52%) of our 54 references were published between 2020 and 2025. Regarding your suggestion that the reference list "may be expanded," we have respectfully re-evaluated our citations. Our approach was to create a comprehensive list that includes both the foundational, older studies in this field and the most current literature. As you observed, almost all of the references we added during our recent revisions are from the last 1-2 years. We believe that the current 54 references adequately cover the scope of our paper by providing a thorough background from the oldest to the newest relevant studies. Therefore, we feel that expanding it further may not be necessary at this time. Thank you for your consideration.

Reviewer 3 Report

Comments and Suggestions for Authors

Thank you for having attempted to address all the issues raised.

Author Response

Open Review (x) I would not like to sign my review report
( ) I would like to sign my review report Quality of English Language ( ) The English could be improved to more clearly express the research.
(x) The English is fine and does not require any improvement.            
  Yes Can be improved Must be improved Not applicable
Does the introduction provide sufficient background and include all relevant references? (x) ( ) ( ) ( )
Is the research design appropriate? ( ) (x) ( ) ( )
Are the methods adequately described? ( ) (x) ( ) ( )
Are the results clearly presented? ( ) (x) ( ) ( )
Are the conclusions supported by the results? ( ) (x) ( ) ( )
Are all figures and tables clear and well-presented? ( ) (x) ( ) ( )
    Comments and Suggestions for Authors

Thank you for having attempted to address all the issues raised.

Dear Reviewer,

Thank you very much for your positive feedback and your valuable, constructive comment.

Your guidance throughout the process of strengthening our manuscript has been invaluable to us.

Sincerely,
